# Evaluating the ability of the operational High Resolution Rapid Refresh model version 3 (HRRRv3) and version 4 (HRRRv4) to forecast wind ramp events in the US Great Plains

Laura Bianco[1,2], Reagan Mendeke[3], Jakob Lindblom[4], Irina V. Djalalova[1,2], David D. Turner[5], and James M. Wilczak[2]

[1]CIRES, University of Colorado, Boulder, CO, USA, 80305
[2]NOAA, Physical Sciences Laboratory, Boulder, CO, USA, 80305
[3]University of Oklahoma, Norman, OK, USA, 73019
[4]Olympia, WA, USA, 98501
[5]NOAA, Global System Laboratory, Boulder, CO, USA, 80305

*Correspondence to*: Laura Bianco (Laura.Bianco@noaa.gov)

**Abstract.** Incorporating more renewable energy into the electric grid is an important part of the strategy to expand our energy portfolio. To make the incorporation of renewable energy into the grid more efficient and reliable, numerical weather prediction models need to be able to predict the intrinsic nature of weather-dependent renewable energy resources. This allows grid operators to plan accurately the amount of energy they will need from each source (e.g., wind, solar, fossil fuel, etc.). For this reason, wind ramp events (rapid changes in wind speed over short periods of time) are important to forecast accurately. This is because one of their consequences is that wind energy could quickly be available in abundance or temporarily cease to exist. In this study, the ability of the operational High Resolution Rapid Refresh numerical weather prediction model to forecast wind ramp events is assessed in its two most recent versions: version 3 (HRRRv3, operational from August 2018 to December 2020) and version 4 (HRRRv4, operational from December 2020 onward). The datasets used in this analysis were collected in the United States Great Plains, an area with a large amount of installed electricity generation from wind. The results are investigated from both annual and seasonal perspectives and show that the HRRRv4 is more accurate at forecasting wind ramp events compared to HRRRv3. Specifically, the HRRRv4 shows increased correlation coefficient and reduced root mean square error relative to the change in wind power capacity factor found in the observations, and in the skill of forecasting both up and down wind ramp events, with a marked increase in the HRRRv4's skill at detecting up ramps during the summer (the HRRRv4 is nearly 50% more skilful than the HRRRv3). This demonstrates that the HRRR's continuing evolution will better support the integration of wind energy into the electric grid.

## 1 Introduction

Many nations are making more investments in renewable energy sources (e.g., hydro, solar, and wind power). This is both to grow their energy portfolio and for economic reasons, given that renewable energy generation does not require the purchase of fuel. According to the International Energy Agency (IEA; Renewables, 2023) more than 500 GW of renewable electricity were added to grids around the world in 2023. This was the largest jump (nearly 50% from the year 2022) in the last two decades. Solar power is taking the lead in this new generation, followed by onshore and offshore wind energy (IEA; Renewables, 2023). Adding into consideration the decreasing costs for wind and solar photovoltaic systems, the IEA report estimates that wind and solar together will account for over 90% of the renewable power capacity that is added over the next five years (to 2028).

Due to the inherent variability of weather-dependent renewable energy resources, numerical weather prediction (NWP) model developers are also investing resources to improve forecasting of the meteorological variables of interest for grid operators, who rely on NWP model forecasts to plan for energy source allocation. Indeed, NWP forecasts of wind speed have been used for over a decade in the decision making associated with integrating wind-generated power into the electrical grid (e.g., Yu et al. 2014; Dong et al. 2016; Jacondino et al. 2021). In this perspective, a series of Wind Forecast Improvement Projects (WFIP) have taken place in the United States (US). These projects have been sponsored by the US Department of Energy (DOE) and the National Oceanic and Atmospheric Administration (NOAA) and included partners from public and private institutions.

The first WFIP (WFIP1; Wilczak et al., 2014, 2015) focused on measuring the impact of including additional meteorological information to the initialization of operational weather prediction models. WFIP1 conducted a 12-month field campaign in 2011-2012 in the US Great Plains, an area of large wind energy production. The second WFIP (WFIP2; Shaw et al. 2019, Wilczak et al. 2019a, and Olson et al. 2019a) focused on an 18-month field campaign that took place in 2015-2017 in the US Pacific Northwest, also an area of large wind energy production. The goal of WFIP2 was to improve physical parameterizations within operational weather prediction models in complex terrain, where the wind flow is modulated by terrain features that are more difficult to simulate. The third WFIP (WFIP3) includes an 18-month field campaign off the coast of New England in the Eastern US, where many offshore wind plants are currently being erected. This ongoing effort, which started in February 2024, aims at supporting offshore wind generation through better forecasting for existing, new, and planned wind farms placed offshore of this area.

All the findings from the WFIP efforts have been transferred to operational versions of the High Resolution Rapid Refresh (HRRR) model. The HRRR is a regional, rapid-refresh, convective-allowing (3 km horizontal grid) NWP model run operationally by the National Weather Service (NWS). The HRRR utilises the Weather Research and Forecasting (WRF) model (Skamarock and Klemp, 2008), wherein the development focused on improving the suite of physical parameterizations and data assimilation scheme to work well with each other for a range of operational forecasting applications. The HRRR first became operational in 2014, and remains as a key forecasting tool used by the NWS and other groups due to its hourly update and high resolution. Details on the HRRR's configuration, data assimilation system, physical parameterizations, and evaluation

can be found in Dowell et al. (2022) and James et al. (2022). This paper will focus on two versions of the HRRR: version 3 (which was operational in the NWS from 12 July 2018 to 1 Dec 2020) and version 4 (which became operational in the NWS on 2 Dec 2020). The primary differences between these two versions are (a) the improved horizontal resolution of the data assimilation system, (b) improved treatment of clouds that are smaller than the resolution of the model, (c) the introduction of wildfire smoke into the model, including its impact on solar radiation, (d) the improvement of the vertical advection scheme, and (e) the reduction in the strength of the numerical diffusion used within the model (Dowell et al., 2022).

The intrinsic variability of the wind is amplified when the wind speed is converted into power, due to the relationship between wind speed and wind power capacity factor. In the range of wind speed values between the cut-in (minimum wind speed below which no power production is obtained by the wind turbines) and cut-off (maximum wind speed above which wind turbines have to be shut down to avoid exceeding turbine design loads) thresholds, a change of a few m s$^{-1}$ in wind speed can result in a change in wind power production of more than 50%. When these large power production changes happen over a short period of time (i.e., less than a couple hours), they are referred to as wind ramps. The accurate forecast of wind ramps is very important for wind energy operators and has potentially large economic impacts, as they need to plan in advance what source of energy will be available to the grid (Jeon et al., 2022; Jin et al., 2024). Turner et al. (2022) and Jeon et al. (2022) already demonstrated that improvements in the operational HRRR have resulted in significant economic savings for the US through better grid operators' decision-making. In their studies, they found appreciable economic gain between HRRR versions 1 (HRRRv1) and 2 (HRRRv2) and a smaller but still appreciable gain between versions 2 (HRRRv2) and 3 (HRRRv3).

The accuracy of the NWP model at forecasting wind ramp events cannot be estimated using standard statistical metrics (e.g., mean absolute error, correlation coefficient, or root mean square error) because these would also take into consideration the periods of time when the wind power is at its minimum or full capacity. Therefore, a tool called the Ramp Tool and Metric (RT&M) was developed to evaluate an NWP model only for the times when wind ramps occur, with the aim of measuring the skill of the NWP model at forecasting wind ramp events (Bianco et al., 2016). The RT&M has been used during WFIP1 (Bianco et al., 2016; Akish et al., 2019) and WFIP2 (Djalalova et al. 2020) campaigns to estimate the improvement in the operational NWP models.

In this study, the RT&M is used to estimate the skill of the operational HRRR model in its two most recent versions, version 3 (HRRRv3) and version 4 (HRRRv4). The analysis is performed using the datasets collected in the US Great Plains, where wind energy production is abundant, and is achieved on an annual basis, as well as on a seasonal basis.

The manuscript is organized as follows: the wind ramp definition and the RT&M used to evaluate the model forecast skill are described in Sec. 2; the area of investigation and the datasets (observational and model) used are presented in Sec. 3; the diurnal and seasonal variability of wind speed and ramp events in the study area are presented in Sec. 4; the skill of the HRRRv3 and HRRRv4 models at forecasting ramp events both from an annual and a seasonal perspective is discussed in Sec. 5. Finally, the summary and conclusions are in Sec. 6.

## 2 Wind ramps definition and description of the RT&M

Weather-dependent energy is subject to rapid changes of power availability over short periods in time, referred to as ramps. In this study, the dependence of wind power capacity factor (P) to wind speed (WS), in the range of wind speed values between 3-16 m s$^{-1}$ (region II of the wind speed to wind power capacity factor curve), is assumed to be given by the formula presented in Wilczak et al. (2019b). This formula is computed using the average of several wind power capacity factor curves for IEC Class 2 turbines.

Additional information to be considered is: (a) below the cut-in wind speed (3 m s$^{-1}$) the wind is insufficient to produce power by the wind turbines, therefore P = 0 (region I of the wind speed to wind power capacity factor curve); (b) between 16 m s$^{-1}$ and the cut-off wind speed (25 m s$^{-1}$) the wind power capacity factor is at its maximum (P = 1, region III of the wind speed to wind power capacity factor curve); and (c) above the cut-off wind speed the wind turbines have to be shut down to avoid strain on the rotor, therefore P = 0 (region IV of the wind speed to wind power capacity factor curve).

The wind speed to wind power capacity factor curve is presented in Fig. 1

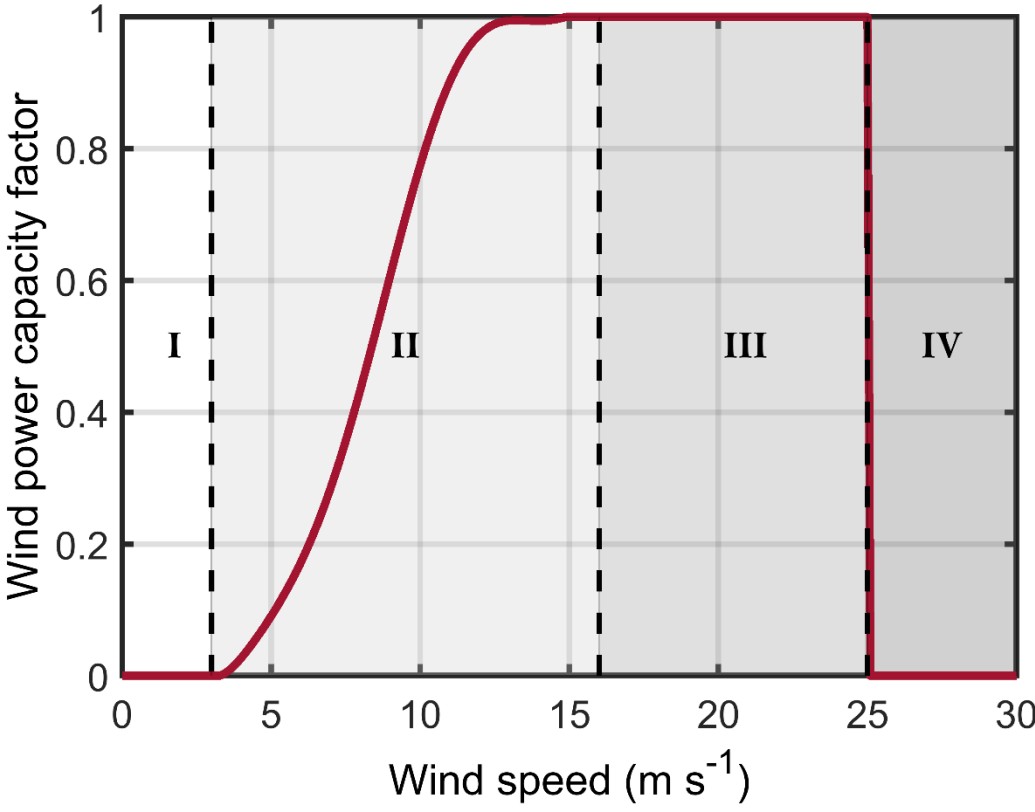

**Figure 1: Wind speed to wind power capacity factor conversion curve. Cut-in wind speed is 3 m s$^{-1}$ and cut-off wind speed is 25 m s$^{-1}$. Regions I, II, III, and IV of the curve are indicated in between the dashed lines.**

The RT&M has three components: the first is the identification of ramp events in the time series of the observed and model power data; the second is matching observed ramp events with those predicted by the forecast model; the final component is scoring the ability of the model to forecast ramp events (both timing and intensity). As an exact definition of a ramp is not unique (i.e., how much the wind power capacity factor has to change and over what time period for the event to be considered a ramp), a metric that is aimed at evaluating an NWP model at forecasting ramp events has to include a range of ramp parameters. Additionally, the skill of a model at forecasting the occurrence of these events has to consider the capability of the model to predict the time of the event (or its central time, Ct), its duration ($\Delta T$), and the amplitude of the change in the wind power capacity factor ($\Delta P$). The RT&M was developed to take into consideration the fact that a ramp is not uniquely defined (several $\Delta P$ and $\Delta T$ combinations have to be considered) and that the skill of the model is a function of accurately forecasting all three Ct, $\Delta T$, and $\Delta P$ variables. This RT&M is described in Bianco et al. (2016).

Equations for the computation of the model skill score at forecasting wind ramp events are formulated for different matching scenarios between forecasted and observed ramps. Specifically, 8 possible scenarios of model vs observed events are considered, consisting of: up/up, up/null, up/down, null/up, null/down, down/up, down/null, down/down, resulting in the 3x3 contingency table except null/null events that do not impact the score. For null scenarios (up/null, null/up, null/down, and down null), the score will be equal to 0. For the nonnull scenarios the score is computed as a cube-root equation dependent on the three nondimensional errors associated with the amplitude, timing, and duration of the ramp, with coefficients based on the 8 different scenarios, as described in detail by Eq. 1-8 of Bianco et al. (2016).

This metric has potential usefulness for grid operators that need to quantify the reliability of NWP models they depend on for their decision making, or for NWP model developers to test whether their efforts at improving the operational model are reflected in better forecasts that can benefit the energy sector.

**3 Area of investigation and dataset description**

According to Table 1.14.B of the US Energy Information Administration (EIA) electric power monthly report (US EIA, 2024), the six states with the most electricity generation from wind in 2023 were Texas, Iowa, Oklahoma, Kansas, Illinois, and New Mexico. These six states combined produced about 64% of total US wind electricity generation in 2023.

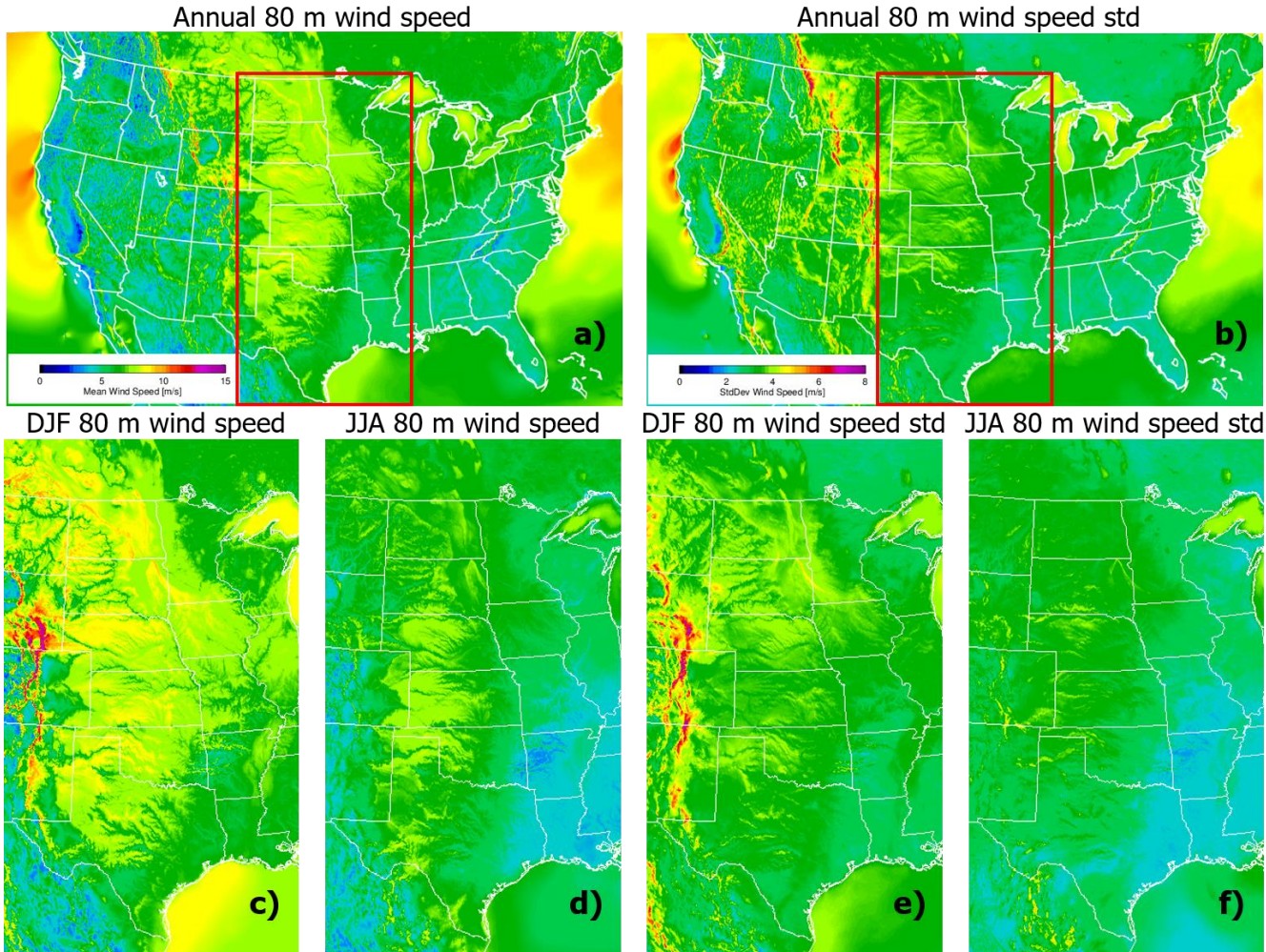

Annual 80 m wind speed | Annual 80 m wind speed std

DJF 80 m wind speed | JJA 80 m wind speed | DJF 80 m wind speed std | JJA 80 m wind speed std

137

**Figure 2: Annual mean (a) and standard deviation (b) of the wind speed at 80 m derived from 1-h forecasts from the HRRR over 2020–2022. Panels (c) and (d) show the mean wind speed for DJF and JJA, respectively, and panels (e) and (f) show the standard deviation of the wind speed for DJF and JJA, respectively (using the same colour bar ranges as in panels (a) and (b)).**

This information is also confirmed by the 2-dimensional wind speed field output at 80 m above ground level (agl) of the HRRR model (Fig. 2), which is a typical height used for wind energy investigations. From this figure, larger values of 80 m wind speed can be seen in the six states listed above, which explains why this region is important for wind energy production (onshore wind turbine locations can be seen here: https://energy.usgs.gov/uswtdb/viewer/#4.55/39.88/-94.56). Another interesting feature shown in Fig. 2 is the lower values of summer 80 m wind speed (Fig. 2d), compared to winter (Fig. 2 c). This will also be explored later in the manuscript when comparing the model to the observations (Section 4).

One of the atmospheric phenomena experienced in the US Great Plains, and of large interest for wind energy, are low-level-jets (LLJs). LLJs have been studied for many years (e.g., Bonner, 1968, Whiteman et al. 1997, Banta et al. 2002, Banta et al., 2008) and occur often in the US Great Plains, particularly in the southern part of it (Freedman et al., 2008). They happen over

relatively flat terrain, during nighttime when the boundary layer is stable, as the ground cools down during the evening boundary layer transition and the flow is decoupled just above the surface. This decoupling leads to an acceleration of the flow above the atmospheric surface layer and produces a layer of air with high-momentum, which often exhibits a maximum in the vertical profile of the horizontal wind. Whiteman et al. (1997) analyzed the climatology of the LLJ in the United States Great Plains from 2 years of radiosonde data and found that the height of the jet maximum occurs most frequently in the 300–600-m height range, with a peak between 300 and 400 m. Of course, it would be ideal in this analysis to use a dataset of wind speeds at hub-height. Unfortunately, this is not possible as there were very few such observational datasets available to carry out a meaningful geographical investigation.

Previous studies (Schwartz and Elliott, 2005; Newman and Klein, 2014) also recognize the fact that, although the wind speed at hub height is the one of interest for wind energy application, most wind speed measurements are taken at 10 m agl as tall meteorological towers are expensive to build, operate, and maintain. Newman and Klein (2014) used the Oklahoma Mesonet surface observation stations and compared the most widely used extrapolation method to relate 10 m measurements to 80 m wind speeds collected by tall towers. They found that the power law, which relies only on the information of wind speed at a reference height (i.e., 10 m agl) and a shear exponent (dependent on atmospheric stability regimes), produced accurate 80 m wind speed estimates from 10 m wind speed observations and concluded that these could be therefore used for increasing our knowledge of hub-height wind speed climatologies.

To ensure that the conclusions of our study are of interest for the wind energy community, we investigate if the results found using 10 m wind speed are applicable to the wind speed field at a typical hub-height, such as 80 m agl. Ramp events can be divided into those that occur because of the strong diurnal variability within the boundary layer, and those that are associated with meteorological phenomena such as cold fronts, gust fronts, or other changes in forcing from transient mesoscale pressure gradient fields. Although the diurnal variation of wind speeds at 10 m and at several 100 m can be out-of-phase (with 10 m wind speeds decreasing during the night time hours while at 300-400 m they may increase at night due to the low-level jet) diurnal variations at both heights are driven by surface and boundary layer fluxes and turbulent mixing. If improvements to the model's parameterization of those diurnal processes increases forecast skill at 10 m, one could speculate that improvements to forecast skill would also be found at greater heights within the boundary layer. Although we only use 10 m observations in our analysis, evaluation of 10 and 80 m winds in the model indicate that improvements to 80 m wind forecasts are in fact expected. The results of this investigation are presented in Appendix A, supporting that our findings can be considered representative of the wind speed atmospheric field of interest for renewable energy and we will thereafter use wind speed observations made at 10 m agl. This study focuses on the geographical area of the US Great Plains, where a large number of observations is available. Model output at the same height will be used for comparison.

**3.1 Observational dataset description and preparation**

The observational dataset used in this study is obtained by the METeorological Aerodrome Reports (METARs) stations, a network of weather stations located mainly in airports and used for flight planning and weather forecasting

(https://aviationweather.gov/data/metar/). The United States Geological Survey (USGS) Wind Turbine database
(https://eerscmap.usgs.gov/uswtdb/) was used to identify the location of the wind turbines. The 10 m agl wind speed
observations at locations that are within 20 km of a wind turbine are extracted. Native METAR data are typically 15-min or
20-min resolution; as the output from the HRRR is hourly, we have linearly interpolated the METAR observations in time to
the HRRR output times (i.e., the top of each hour). Generally, the observation close to the top of the hour is within 10 minutes.
Fig. 3 shows the geographical location of the METAR weather stations used in this study, which are superimposed over the
topography of the study area. The location of the METAR weather stations allows for a geographically well distributed analysis
of the results.

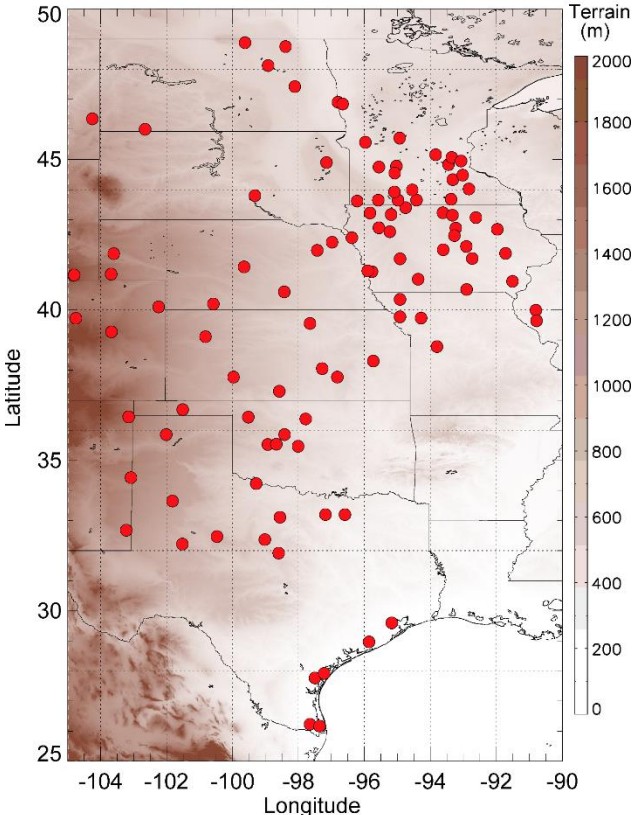


**Figure 3: Geographical location of the METAR weather stations used in this study superimposed on the topography of the study**
**area.**

## 3.2 Operational model description and preparation

As mentioned earlier, the model of interest in this study is the operational HRRR, which uses a 3-km grid spacing. The HRRR is initialized from the operational Rapid Refresh model (RAP; Benjamin et al. 2016), and assimilates other observations (e.g., METAR, AMDAR aircraft, and weather radar data) to derive its analysis, from which forecasts are initiated. The HRRR provides 18 h forecasts every hour, but for four times per day the maximum forecast length is extended. For those four initialization times (00:00, 06:00, 12:00, and 18:00 UTC), the HRRRv3 provides forecast out to 36 forecast hours, while the HRRRv4 goes out to 48 hours. Additional details on the model configurations and parameterizations are provided in Dowell et al. (2022).

The "day-ahead" forecast is particularly useful for the energy community, as that is when decisions are made on the amount of fossil fuel generation to have on-line, which depends on the amount of wind (and solar) energy that is expected to be generated. Thus, we focused on the 00:00 UTC initialization, and used the 13-to-36 h forecasts from both the HRRRv3 and HRRRv4. For each model, the 13-to-36 h forecasts were concatenated to provide continual temporal coverage across the time periods analyzed. However, an artificial "ramp" could be created when merging the 36-h forecast initialized at 00:00 UTC on day X with the 13-h forecast initialized on day X+1 at 00:00 UTC due to a slight bias between the two forecast runs. To reduce this impact, a 3-point (equivalent to 3 hours) smoother was applied at the stitching points of the transition times (point 1 is 36-hr forecast of day X and point 2 is 13-hr forecast of day X+1). For these two points, the model output is an average over 3 points (for point 1 these will be: 35-hr and 36-hr forecasts of day X, and 13-hr forecast of day X+1; for point 2 these will be: 36-hr forecast of day X, and 13-hr and 14-hr forecasts of day X+1) with the two outer points having 25% weight and the central time having a 50% weight.

An example of how the model forecast runs are combined together to provide a time series of wind power capacity factors to compare with the observations is presented in Fig. 4. Both observed and modeled wind power capacity factors are obtained applying the wind power curve to the 10 m observed and modeled wind speeds. In this example, a time series of the observed wind power capacity factors at 10 m agl for the KEWK METAR weather station, located in Kansas, is presented with the black solid line for the time period from 8 April 2021 to 13 April 2021. Dashed lines, in different colors, present the HRRRv4 forecasts (out to 48 forecast hours), at 00Z initialization times each day. The solid red line represents the time series of the model data obtained by the procedure described above. In this example, several ramp events are identifiable. The sharpest down ramp happens at the end of 8 April 2021, while the sharpest up ramp event is noticeable at the end of 9 April 2021. During these events, the available wind power capacity factor for a wind turbine at this location could easily go from its maximum to zero and vice-versa. The HRRRv4 tends to reproduce the wind power capacity factor fairly well, with some inaccuracy in the timing, amplitude, and duration of the ramp events. These inaccuracies are taken into consideration by the RT&M when the skill of the model is computed.

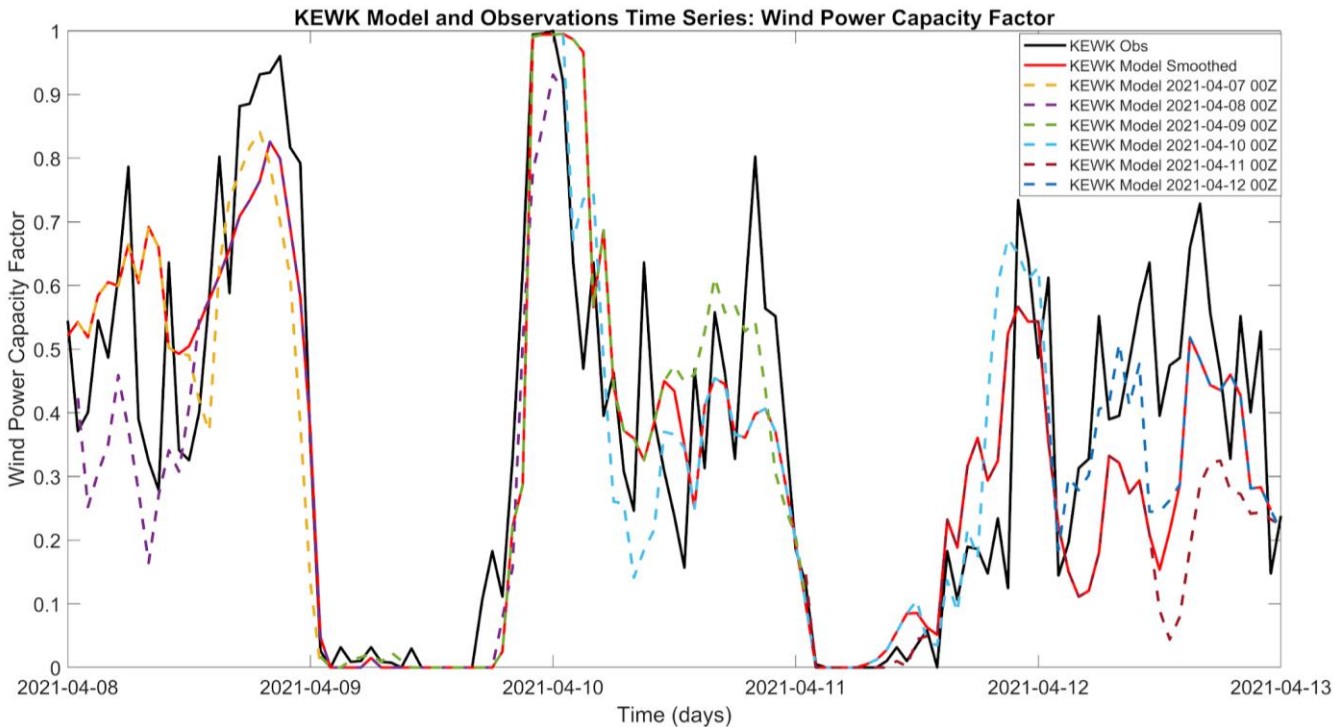

226

**Figure 4: Time series of the wind power capacity factor from 8 April 2021 to 13 April 2021 from the KEWK METAR weather station, located in Kansas (black line), and of the HRRRv4 forecasts (out to 48 forecast hours) at 00Z initialization times (dashed lines in different colour for the different days). The wind power capacity factors are obtained converting the 10 m observed and modelled wind speeds.**

An optimal way to evaluate the relative skill of the HRRRv3 against the HRRRv4 would be to use periods of time when both models are available. However, since we are assessing the operational models, there are no periods of overlap that can be used. To prove that using different time periods for the two versions of the HRRR is a valid alternative, we looked at the geographical distributions of wind ramp events found on the 10 m agl wind power capacity factor of the HRRRv3 in 2020 and the HRRRv4 in 2021 and 2022. Fig. 5 shows the number of ramp events (for the type of ramps defined as having a $\Delta P/\Delta T \geq 40\%/2hrs$) at each of the observational locations, represented with colored circles function of the number of identified ramps. The geographical distribution of the number of wind ramp events agrees with the annual wind speed geographical distribution presented in Fig.2. Additionally, the geographical distribution of the number of these events are very similar between HRRRv3 in 2020 (panel a), HRRRv4 in 2021 (panel b), and HRRRv4 in 2022 (panel c). Of course, it has to be considered that the inter-annual variability of the wind distribution across the study area could impact the results of this study. A discussion about this possibility is included in Appendix B. It is interesting to note how for all three years the number of ramps is larger in the west side of the study area, in the north-western part of Texas, in the southeast locations closer to the Gulf of Mexico, and in Oklahoma. Consistently between the years, there are fewer ramps in the central part of Texas and on the eastern side of the

study domain. The central, northern, and north-eastern parts of the study area experience fewer ramp events, and the numbers
are relatively consistent for all three years. This confirms that even though the time periods used to evaluate the HRRRv3 and
HRRRv4 are not coincidental, the comparison is still valuable.

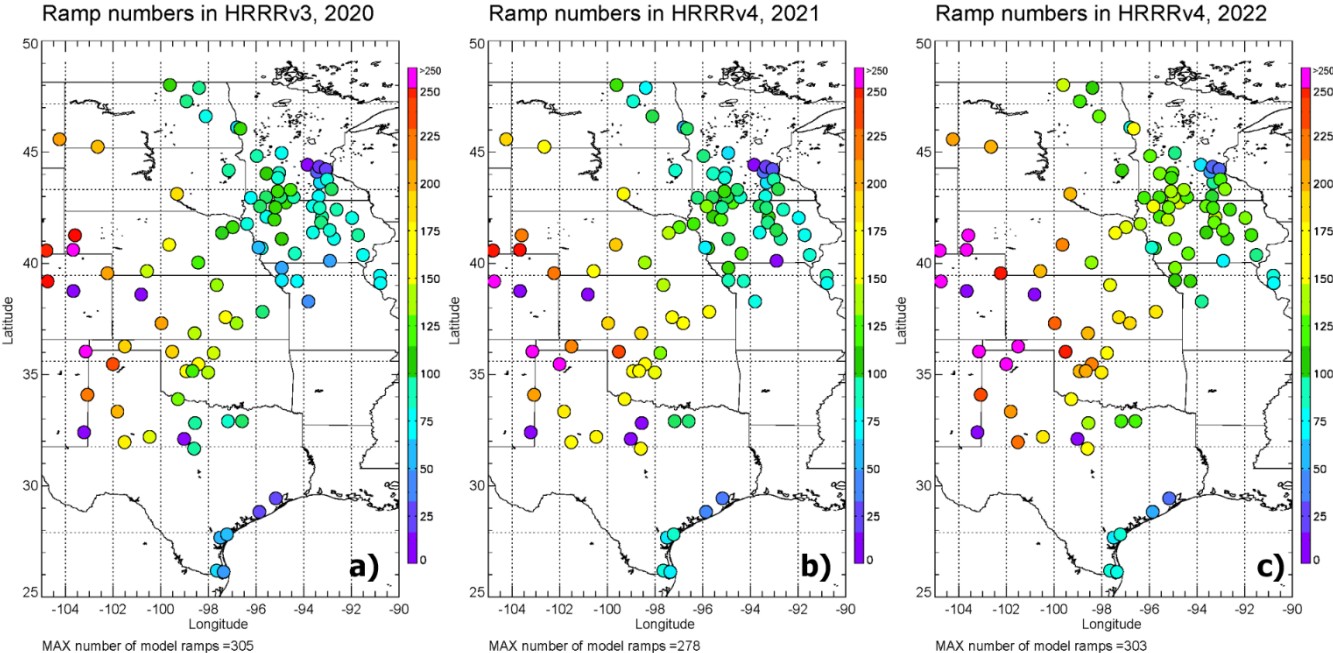


**Figure 5: Geographical distribution of wind ramp events ($\Delta P/\Delta T \geq 40\%/2hrs$), at each tower location, by year: HRRRv3 in 2020 is in panel a, HRRRv4 in 2021 and 2022 are in panel b and c, respectively.**

Similarly, the geographical distribution of the ratio between the number of forecast wind ramps (for the type of ramps defined
as having a $\Delta P/\Delta T \geq 40\%/2hrs$) and those observed, for the three years is presented in Fig. 6 (panels a, b, and c). It is noticeable
how the models tend, in general, to find fewer ramp events (ratio less than 1), which is expected due to the smoother wind
field output of the model compared to observations. This is in accordance with what was found by Bianco et al. (2016) and by
Djalalova et al. (2020). Nevertheless, it is encouraging to find that the average of the ratio over the study area of the ratio tends
to get closer to 1 for the HRRRv4 periods relative to the HRRRv3 period (being equal to $0.53 \pm 0.24$, $0.58 \pm 0.24$, and $0.68 \pm$
$0.22$ respectively for the years 2020, 2021, and 2022).
To further show that the ratio between the number of forecast wind ramps and those observed improves over the years and the
model versions, we present the geographical distribution of the improvement from 2020 to 2021 and from 2020 to 2022, in
panels d and e of Fig. 6, respectively. As noticeable, at most of the stations (72.5% of panel d, and 67% of panel e) the
improvement is positive.

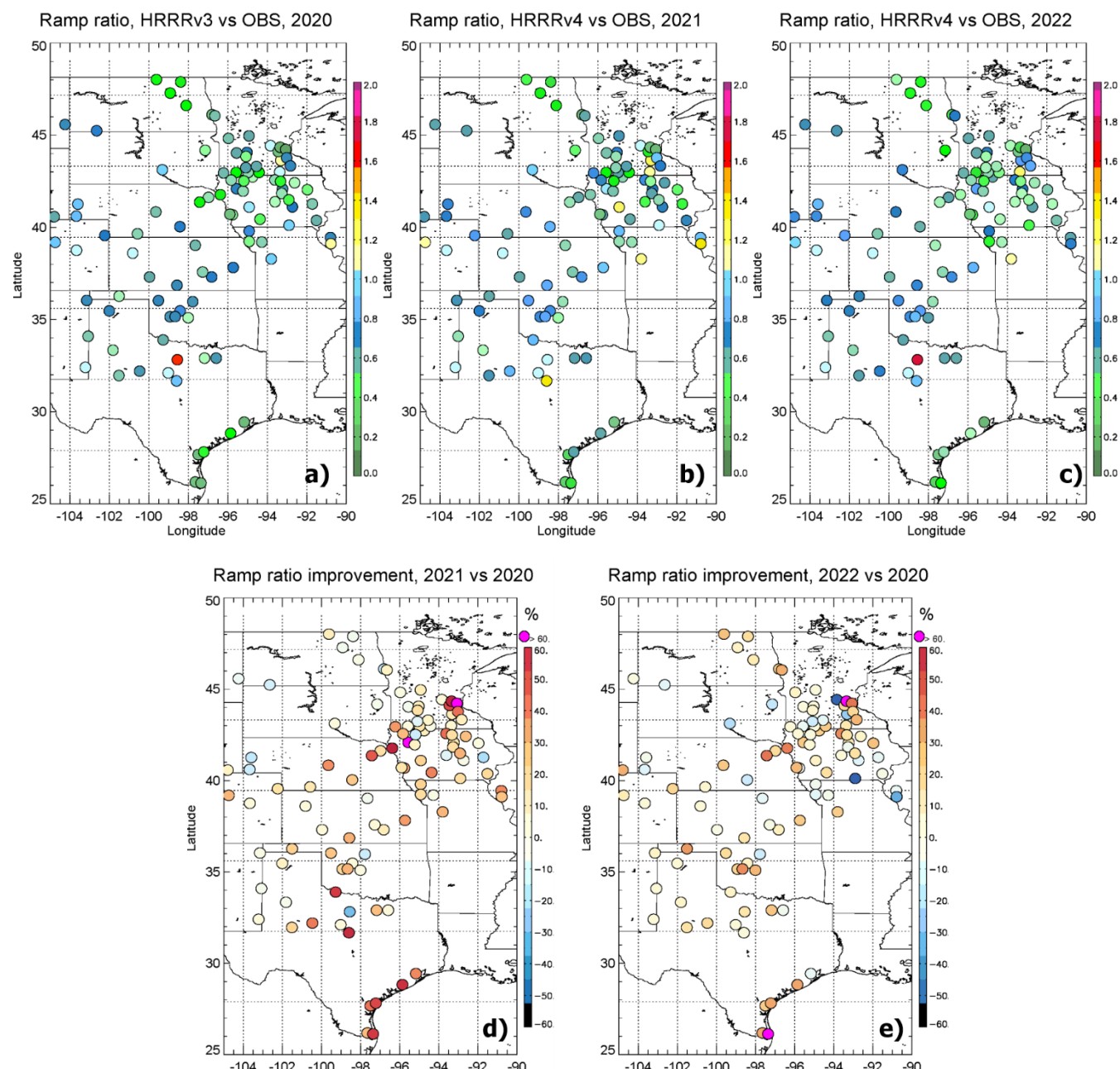

**Figure 6: Geographical distribution of the ratio of the number of model vs observational wind ramp events ($\Delta P/\Delta T \geq 40\%/2hrs$), at each tower location, by year: HRRRv3 in 2020 is in panel a, HRRRv4 in 2021 and 2022 are in panel b and c, respectively). Improvement in this ratio is in panel d for HRRRv4 in 2021 vs HRRRv3 in 2020, and in panel e for HRRRv4 in 2022 vs HRRRv3 in 2020.**

## 4 Diurnal and seasonal variability of 10 m wind speed and ramp events in the observational and model datasets

The composites of the diurnal variability of the 10 m wind speed field over the study area are presented in Fig. 7 (right y-axes), for the four seasons in the different years. The spring, summer, fall, and winter seasons are presented in panels a, b, c, and d for 2020, in panels e, f, g, and h for 2021, and in panels i, j, k, and l for 2022. The mean diurnal observed wind speeds are in blue and modeled values in magenta. The diurnal cycle of the 10 m wind speed field is clearly evident, with winds weaker at night time and increasing in value starting from sunrise into the daytime (local time in the US Great Plains is: $LT = UTC - 5$). The strongest daytime winds are experienced in the spring, while summer has the weakest 10 m wind speeds throughout the whole day. The models are able to reproduce the diurnal variability of this field (magenta and blue time-series for the model and observations, respectively), across the three years and for the different seasons. On the left y-axes are plotted the total number of ramps measured by the observations and by the models, for both up ramps (positive $\Delta P$) and down ramps (negative $\Delta P$).

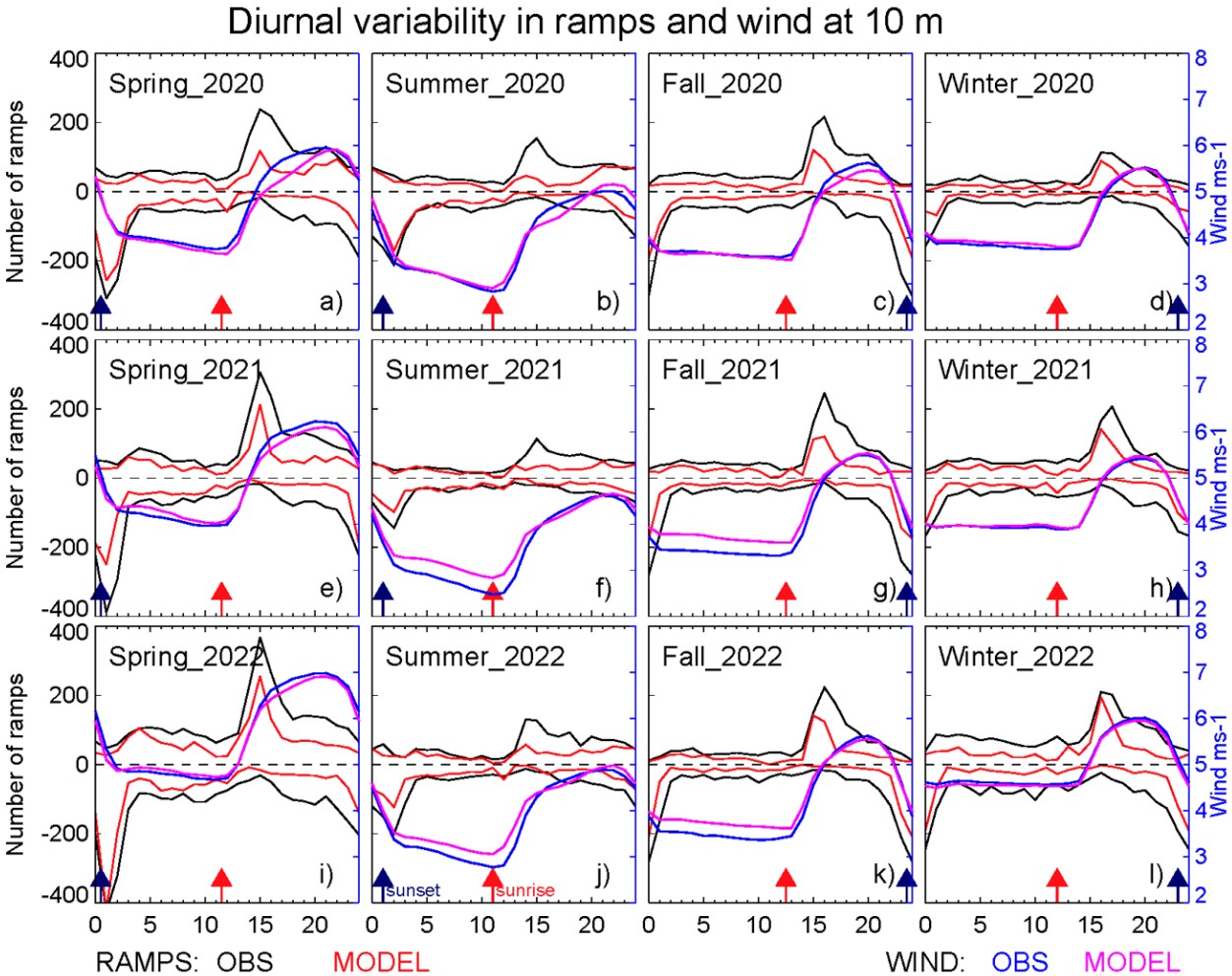

289

**Figure 7: Left axes: Total number of wind ramp events for one ramp definition (ΔP/ΔT ≥ 40%/2hrs) over the study area as a function of time-of-day (hours UTC), for the four seasons. Winter is defined as December, January, and February; spring as March, April, and May; summer as June, July, and August; and fall as September, October, and November (left to right: spring, summer, fall, and winter) in the different years (panels a, b, c, and d: 2020; panels e, f, g, and h: 2021; and panels i, j, k, and l: 2022). Right axes: Composites of the diurnal variability of the 10 m wind speed field over the study area, for the four seasons in the different years. Sunrise and sunset times are denoted by the red and navy arrows, respectively.**

It is apparent that the daily distribution of ramp events analyzed in this study follows the diurnal cycle of the 10 m wind speed for all seasons with down ramps more evident around 22:00-03:00 UTC when the 10 m wind speed sharply decreases (around sunset), and up ramps more evident around 12:00-17:00 UTC when the 10 m wind speed sharply increases (around sunrise). For this reason, the diurnal peaks in the ramps coincide with the largest temporal changes in the mean wind speed. We could speculate that a reverse behavior in the diurnal cycle of wind speed may appear at higher heights, especially at nighttime. This

consideration is particularly valid at the height of the nose of the LLJ although, as mentioned earlier, Whiteman et al. (1997)
found that the height of the jet maximum occurs most frequently between 300–600-m.
Although, as discussed in Fig. 6, the number of observed ramps is in general larger than the number of model ramps, we
performed a statistical analysis for the matched wind ramp events (model and observed ramps are matched when the distance
between their relative central time is less than the defined time window length, i.e. 2hr for the type of ramps defined as having
a ΔP/ΔT ≥ 40%/2hrs). The correlation and root mean square error (RMSE) in ΔP for these matched events at all sites are
presented in Fig. 8. For HRRRv4 we used the averaged correlation coefficient and RMSE of years 2021 and 2022. With the
exception of winter, both the statistical metrics improve in HRRRv4 compared to HRRRv3.

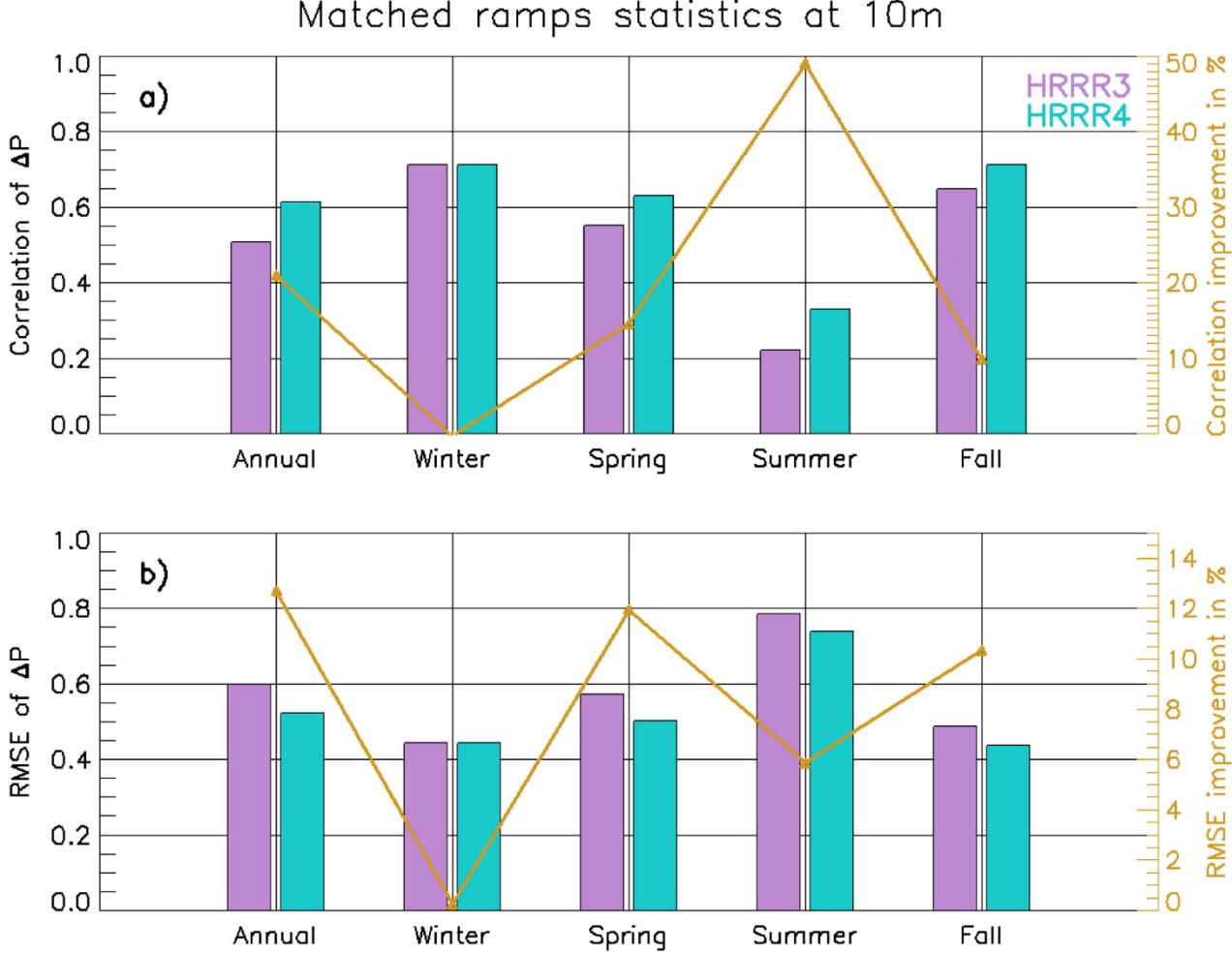

**Figure 8: Left axes: Bar charts of correlation coefficients (panel a) and RMSE (panel b) of observed vs modelled ΔP (for matched**
**wind ramp events defined as ΔP/ΔT ≥ 40%/2hrs) by year (left to right: annually and by season). There are two different sets of data,**

with 2020 in violet and the average of years 2021 and 2022 in aqua. Right axes: Percentage improvements in correlation (panel a), and in RMSE (panel b).

## 5 Models' skill at forecasting ramp events

### 5.1 Annual geographical analysis

In this section, the geographical distribution of the annual improvements in the skill of the HRRRv4 versus HRRRv3 is discussed. The improvement in the skill is computed as:

$$Improvement\ (\%) \ = \ [(Skill\ HRRRv4)\ -\ (Skill\ HRRRv3)]\ /\ (Skill\ HRRRv3)\ x\ 100 \qquad (1)$$

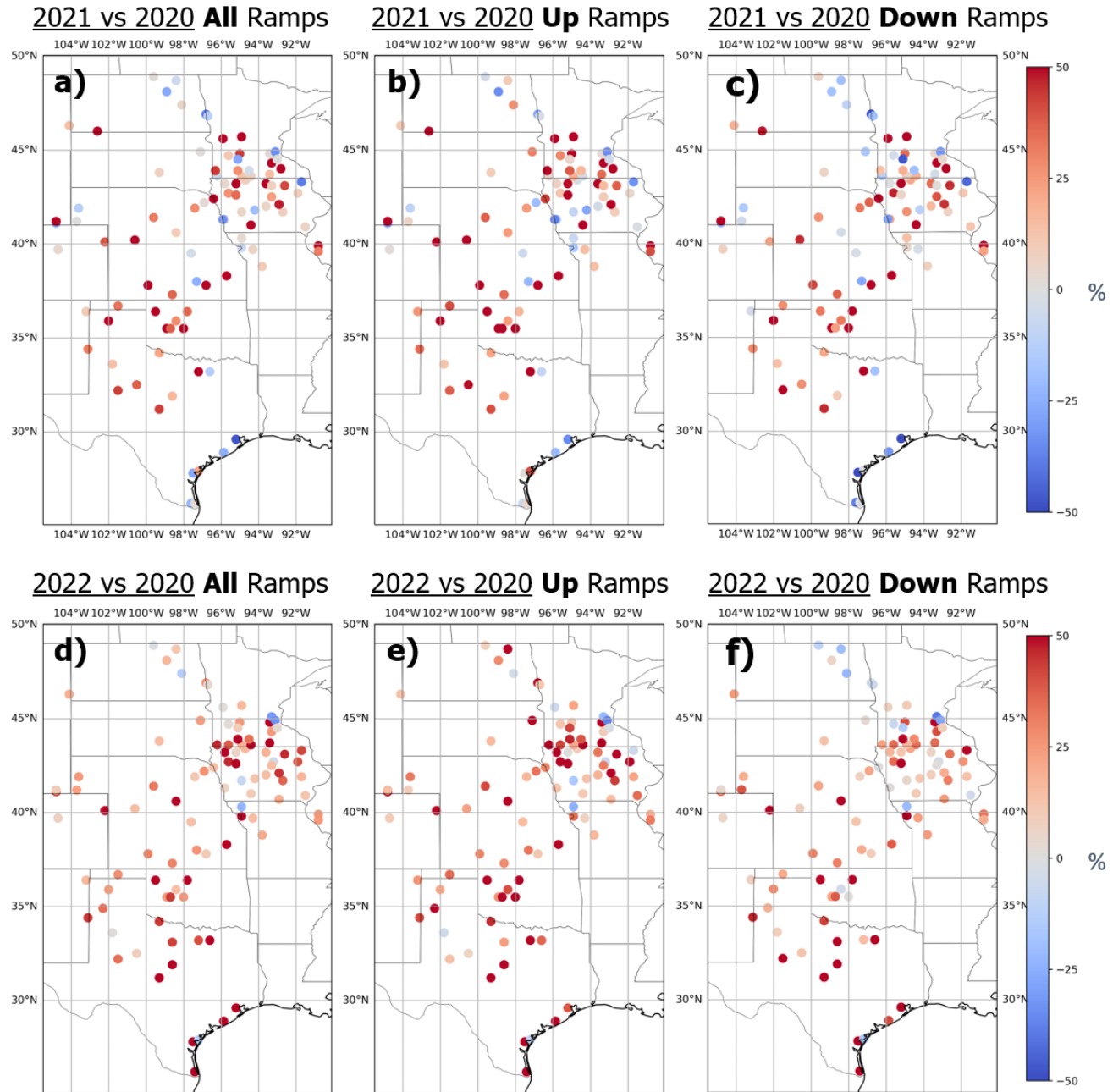

Figure 9: Geographical distribution of the annual improvement of the HRRRv4 vs HRRRv3 skill score at forecasting ramp events at each tower location, by year (panels a, b, and c: 2021 vs 2020; panels d, e, and f: 2022 vs 2020), for all ramps (panels a and d), up ramps (b and e), and down ramps (c and f).

Fig. 9 presents the improvements in red (or degradation in blue) in the skill scores for year 2021 vs 2020 and year 2022 vs
2020, and for all ramps, up ramps only, and down ramps only. The predominance of increased skill (red colours) is apparent
and it is quite uniform spatially, despite the different geographical distribution of wind ramp events seen in Fig. 5, denoting
the improvement found in the HRRRv4 compared to the HRRRv3, confirming that physical developments in HRRRv4 are
valid across the study area. This is true for all ramps, and for up ramps slightly more than for down ramps
**5.2 Annual and seasonal statistical analysis**
A similar analysis to the one presented in the previous sections was repeated for the individual seasons and is presented here
averaged over the study area. The left axes of Fig. 10 presents bar charts with the ramp skill scores averaged by model version
annually and by season, for all ramps, up ramps only, and down ramps only; right axes show the percentage improvements in
skill score annually and by season, for all ramps, up ramps only, and down ramps only.

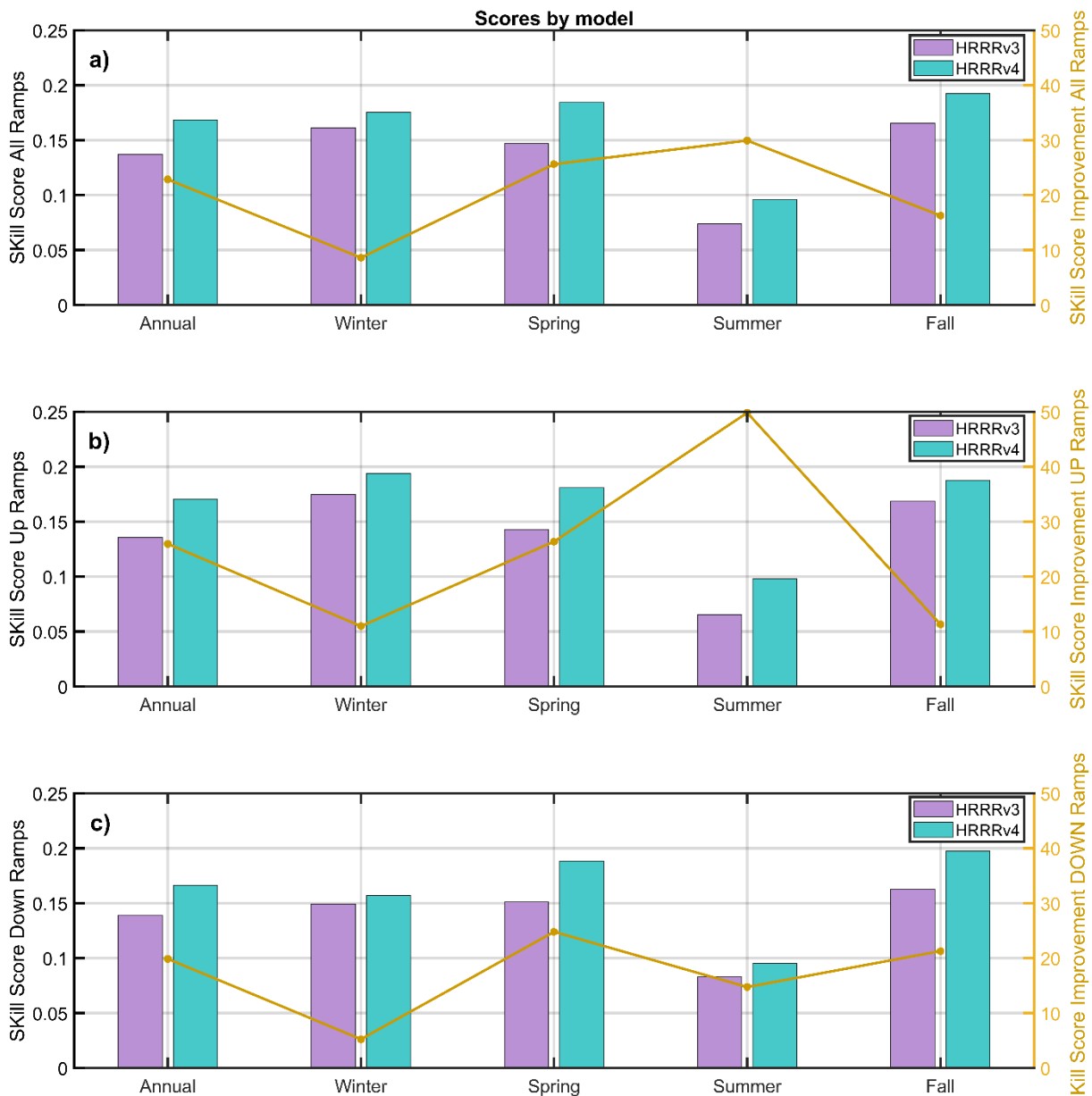


**Figure 10: Left axes: Bar chart with skill scores averaged by model version annually and by season, for all ramps (panel a), up ramps only (panel b), and down ramps only (panel c). Right axes: Percentage improvements in skill score annually and by season, for all ramps (panel a), up ramps only (panel b), and down ramps only (panel c).**

Most noticeable is the marked increase in the skill of detecting up ramps in HRRRv4 during the summer, with HRRRv4 nearly 50% more skilful than HRRRv3. Across all seasons, and for both up ramps and down ramps, the skill of the HRRRv4 is improved relative to that of HRRRv3. Inter-annual variability can play a role in the skill of the model by year; nevertheless, in Appendix B we show that although there is variability in the hub-height wind field between year 2021 and 2022, in both years the skill of the model (HRRRv4) has improved substantially, with respect to that of year 2020 (HRRRv3). For instance, in spring and fall, the increase in skill score is consistently greater than the inter-annual variability as it will be shown in Appendix B, Fig. B2.

**5.3 Daytime and night time statistical analysis**

Since it could be argued that our results are dependent on atmospheric conditions, it would be helpful to know under which conditions conclusions drawn from 10 m data are most robust, and under which conditions further caution is needed.

To see if the improvements presented in the previous section are still consistent between stable vs unstable atmospheric conditions, the dataset was divided into night time and daytime (due to the lack of temperature measurements at different levels from which to determine stability). We then recomputed the models' skills and skill improvements over these different time periods for ramps defined as $\Delta P/\Delta T \geq 40\%/2hrs$.

The daytime period is selected to be 12:00 to 22:00 UTC and the night time is 23:00 UTC plus 00:00 to 11:00 UTC. The results of this exercise showed that the daytime skill of the HRRRv4 years compared to the HRRRv3 year improved by 10.3% and 9.1% in 2021 and 2022, respectively, and that the night time skill of the HRRRv4 years compared to the HRRRv3 year improved by 9.0% and 21.9% in 2021 and 2022, respectively. These results show that, although there are differences in values, the improvements are still consistently positive for both daytime and night time periods, and for both HRRRv4 years, compared to the HRRRv3 year.

**6 Summary and conclusions**

To increase energy availability and meet the demands for new electricity generation, many nations are investing in renewable energy resources. Since the availability of renewable energy resources is inherently weather-dependent, numerical weather prediction (NWP) model developers are also investing resources to improve the forecast of the meteorological variables of interest for grid operators.

In this study, the operational High Resolution Rapid Refresh (HRRR) numerical weather prediction model is assessed in its ability to forecast wind ramp events. Wind ramp events are rapid changes in wind speed over short periods of time and their accurate forecast is very important for wind energy operators, so that they can reliably plan what source of energy to count on for the grid. The two most recent versions of the HRRR are considered in this study: version 3 (HRRRv3, operational from August 2018 to December 2020) and version 4 (HRRRv4, operational from December 2020 onward). Datasets used in this analysis were collected in the United States Great Plains, an area with a large amount of installed electricity generation from

wind. This study uses wind speed observations from METeorological Aerodrome Reports (METARs) stations made at 10 m agl, and model output at the same height. Our analysis of 10 and 80 m (a typical hub-height) winds in the model indicate that improvements to 80 m wind forecasts are in fact expected in HRRRv4 compared to HRRRv3. We also found that the number of ramps at 10 m correlates well with those at 80 m (R = 0.82 for up ramps and R = 0.84 for down ramps), but we recognize that a correlation of 0.84 explains only 70% of the variance between 10 and 80 m wind speeds and number of ramps at those two heights. The remaining 30% are uncertainties that could possibly reflect in different diurnal wind speed and ramp events behaviours at these two heights.

The evaluation of the HRRR model in its two versions is performed using the Ramp Tool and Metric (RT&M), a tool aimed at measuring the skill of an NWP model at forecasting wind ramp events. This tool takes into consideration the fact that a ramp is not uniquely defined and measures the capability of a NWP model to accurately forecast the time of the event, its duration, and the amplitude of the change in the wind power capacity factor.

The results are investigated from both annual and seasonal perspectives and show how the HRRRv4 is more accurate at forecasting wind ramp events compared to HRRRv3. The HRRRv4 demonstrated notable improvements in the skill of forecasting wind ramp events, compared to the skill of HRRRv3, with increased correlation coefficient and reduced root mean square error relative to change in wind power capacity factor found in the observations. Importantly, this analysis shows that across all seasons, and for both up and down ramp events, the skill of the HRRRv4 is improved relative to that of HRRRv3, with a marked increase in the HRRRv4's skill at detecting up ramps during the summer (HRRRv4 nearly 50% more skilful than HRRRv3). Some of the advances between the versions of the model that likely contributed to the improvements found in this study are: improved higher-resolution data assimilation system, which provides better detailed initial conditions for the model; reduction in the solar radiation bias at the surface that is the result of the improved treatment of clouds, as the net radiation at the surface drives the surface energy budget which itself helps to drive turbulent mixing in the boundary layer; and the reduction of the diffusion terms in the model, which allows for finer scale features to be maintained longer into the forecast before they dissipate.

This study demonstrates the positive evolution of the operational HRRR model to support the integration of wind energy into the electric grid.

**Appendix A**

To demonstrate that the results of our study are of interest for the wind energy community, we investigate representativeness of 10 m wind speed to 80 m wind speed. As a first step, we compared the HRRR model output at 2 levels: 10 m and 80 m agl over the time period from 2020-2022. We found a correlation coefficient equal to 0.84 between wind speed values at these 2 heights. In addition, we converted the time series of the model wind at these levels to power and identified the number of ramps that reached 40%/2hr at both levels. In Fig A1 we show the total number of ramps at each METAR weather station location. In general, we found that the number of ramps at 10 m is around 3 times less than the ramps at 80 m, but the correlation between the number of ramps at these 2 levels over all locations is high (R = 0.82 for up ramps and R = 0.84 for down ramps).

We recognize that a correlation of 0.84 explains only 70% of the variance between 10 and 80 m wind speeds and number of
ramps at those two heights. The remaining 30% are uncertainties that could possibly reflect in different diurnal wind speed
and ramp events behaviours at these two heights.

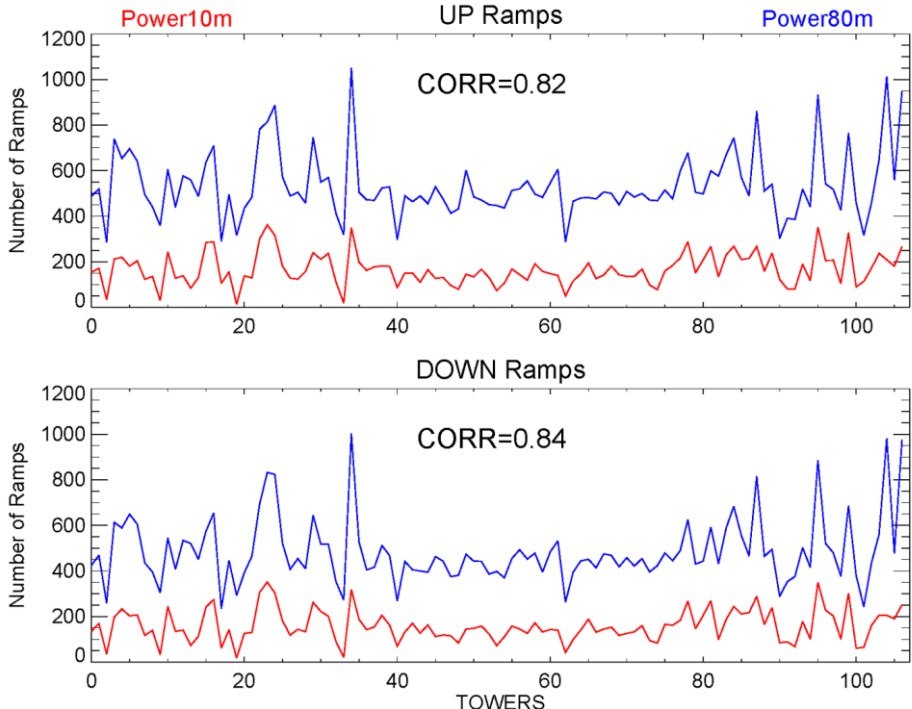


**Figure A1: Total number of ramps (up ramps in upper panel and down ramps in bottom panel) by METAR weather stations for**
**years 2020-2022. Red lines are relative to 10 m wind power capacity factor and blue lines are for 80 m wind power capacity factor.**

We also looked at the geographical distribution of the ramps at these 2 levels, as presented in Fig. A2. The number of ramps
at each site in this figure is normalized by the maximum number of ramps at that level over the entire domain. This
demonstrates that the spatial pattern of the occurrence of wind ramps, both up and down ramps, is qualitatively very similar at
the two heights.

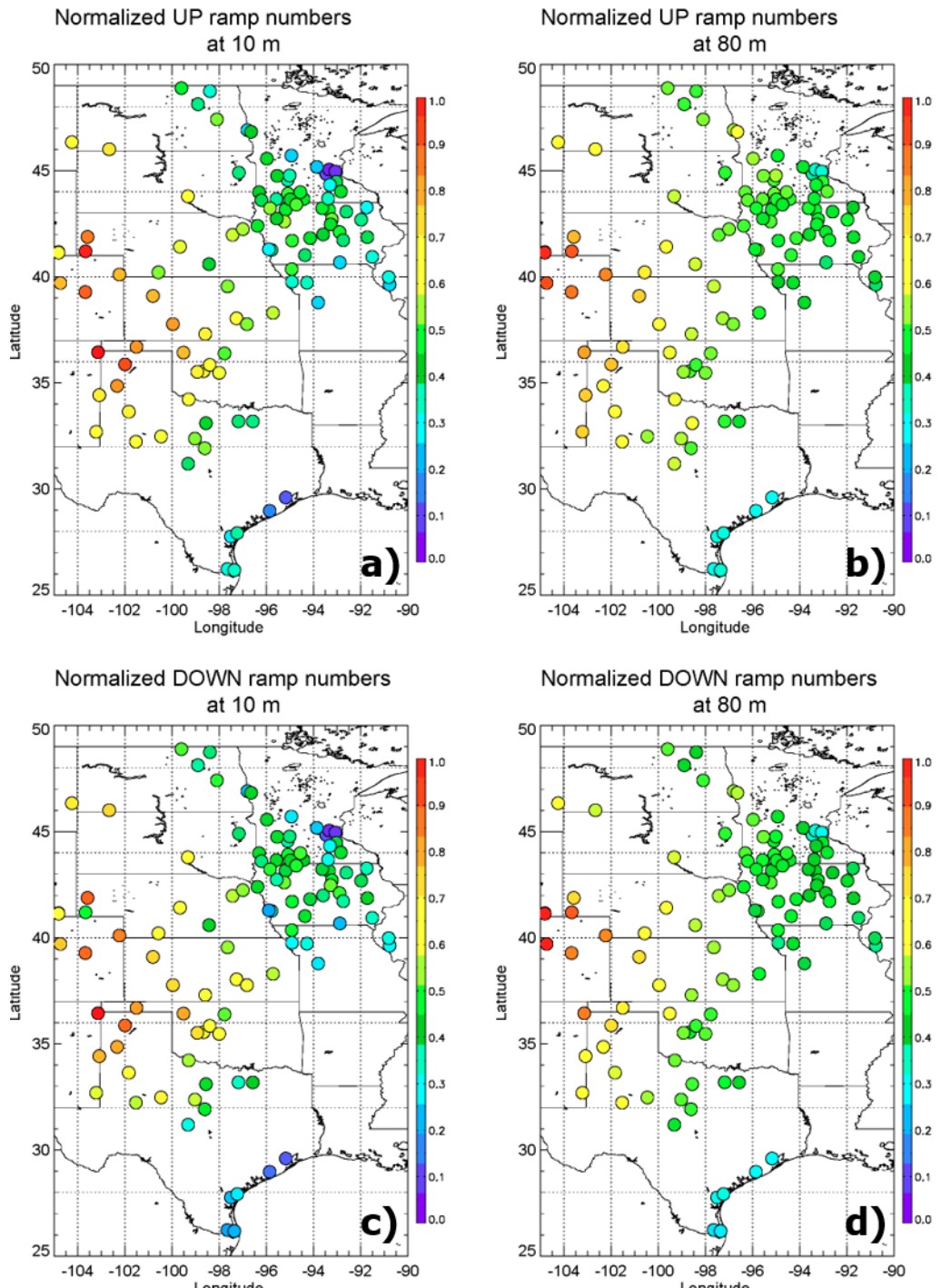


**Figure A2: Normalized number of up ramps (panels a and b) and down ramps (c and d) for wind power capacity factor at 10 m**
**(panels a and c) and at 80 m (panels b and d).**

As noted in the main body of the manuscript, for all three years combined the normalized number of ramps is larger in the
west side of the study area, in the north-western part of Texas, in Oklahoma, and Kansas compared to the north-east part of
the domain. The normalized geographical distribution is consistent between the 10 m and 80 m levels. As it could be expected,
the geographical distribution is smoother at 80 m.
Although 80 m wind speeds are not measured in many locations compared to the availability of METAR stations observations,
we used the long-term routine measurements collected at the Central Site of the ARM Southern Great Plains (SGP)
Observatory in OK (lat: 36.6050 N; lon: -97.4850 W; alt: 318m; Sisterson et al. 2016). At this location routine radiosondes are
launched nominally every 6 hours. The time-height cross section of wind speeds by year is presented in Fig. A3, with
corresponding correlation coefficient values for wind speed and wind power capacity between the 10 m and the levels above.
Of course, this value decreases rapidly with height, but the correlation between the 10 m level and the next few levels is high
($R = 0.94$ for 10 m vs 80 m wind speed, and $R = ~0.8$ for 10 m vs 80 m wind power capacity factor) for all 3 years.

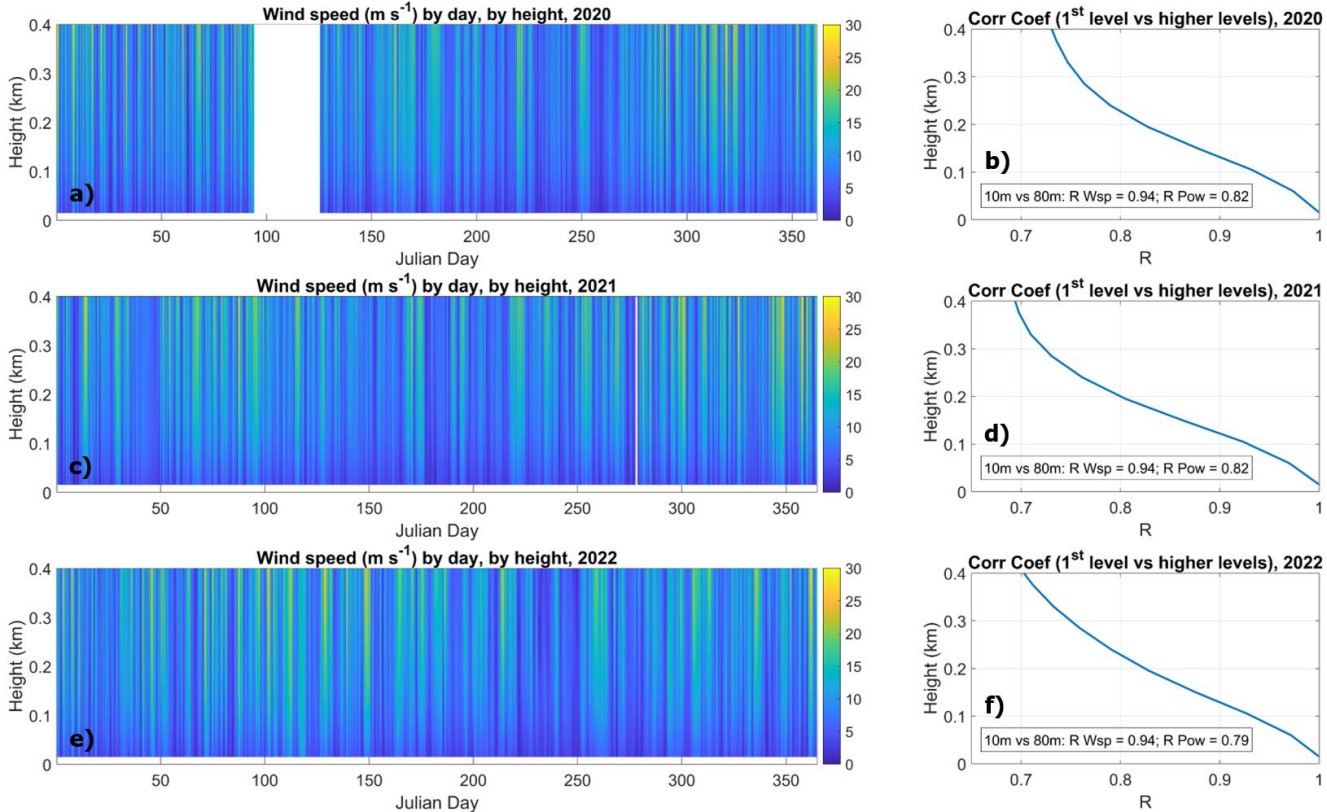


**Figure A3: Time-height cross section of wind speeds by year (2020 in panel a, 2021 in panel c, and 2022 in panel e) at the SPG site.**
**Corresponding profiles of correlation coefficient values for wind speed between 10 m and the levels above are on the right panels**
**(2020 in panel b, 2021 in panel d, and 2022 in panel f). Note that during the 3 April–5 May 2020 period, the SGP site was shut down**
**due to the COVID-19 pandemic.**

Additionally, at this site we computed the correlation between the model and the radiosonde observed winds at 80 m for those three years, finding an improvement in R from 0.85 in 2020 (HRRRv3), to 0.86 in 2021 and 2022 (HRRRv4). We also used high-frequency (10 Hz) observations of wind speed from a sonic anemometer (R3-50, manufactured by Gill Instruments) located on a 60 m tower at the same site. Sonic data were averaged at the top of the hour (plus/minus 5 minutes) providing a more complete dataset compared to the radiosonde one. In this case we found an improvement in R from 0.78 in 2020 (HRRRv3), to 0.79 in 2021 (HRRRv4), to 0.84 in 2022 (HRRRv4) between 80 m model and 60 m sonic wind observations. Furthermore, the comparison with the 60 m sonic observations was repeated dividing the dataset into night time and daytime, similarly to what was presented in Section 5.3. For daytime, correlation coefficient values were found to be equal to 0.84 in 2020 (HRRRv3), to 0.80 in 2021 (HRRRv4), and to 0.87 in 2022 (HRRRv4). For night time, correlation coefficient values were found to be equal to 0.73 in 2020 (HRRRv3), to 0.78 in 2021 (HRRRv4), and to 0.81 in 2022 (HRRRv4). Although this is at one site only, this result aligns with the findings presented in Section 5.3, that in stable conditions the correlation was much improved in HRRRV4 relative to HRRRV3. This supports our speculation that improvements of HRRRv4 compared to HRRRv3 to ramp skill at 10 m would also be found at hub height, although to prove this statement with more certainty, we would need a more appropriate dataset.

**Appendix B**

Inter-annual variability of wind speed in the study area has to be considered as a possible factor impacting the results of this study. We looked at the 2-dimensional wind speed field output at 80 m agl of the HRRR model individually for years 2020, 2021, and 2022, and for winter and summer months, as presented in Fig. B1.

454

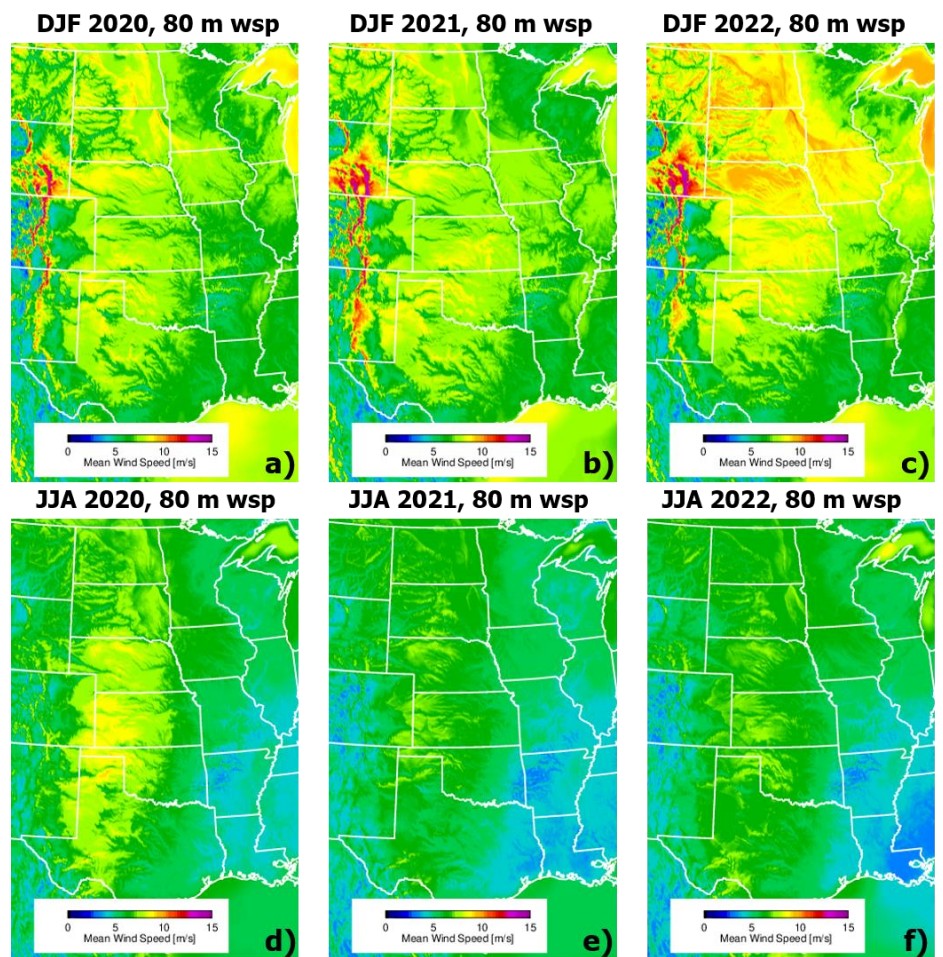

**Figure B1: Winter (DJF; a, b, and c) and summer (JJA; d, e, and f) geographical distribution of the wind speed at 80 m derived from 1-h forecasts of the HRRR over 2020 (a and d), 2021 (b and e), and 2022 (c and f).**

From this figure we do see that 80 m wind speeds are similar in winter months between years 2020 and 2021, but are stronger in 2022, while they are stronger in summer 2020 compared to summer months of 2021 and 2022.

Nevertheless, if we look at the skill score by individual years (Fig. B2), we notice that although there are some differences in skill score between years 2021 and 2022 (with the same HRRRv4 model), the skill score is still improved in both years with HRRRv4 (2021 and 2022), compared to HRRRv3 (2020). This confirms that although inter-annual variability can impact the score of the model (as for example for summer down ramps, Fig. B2 panel c), HRRRv4 is still doing better capturing wind ramps than HRRRv3.

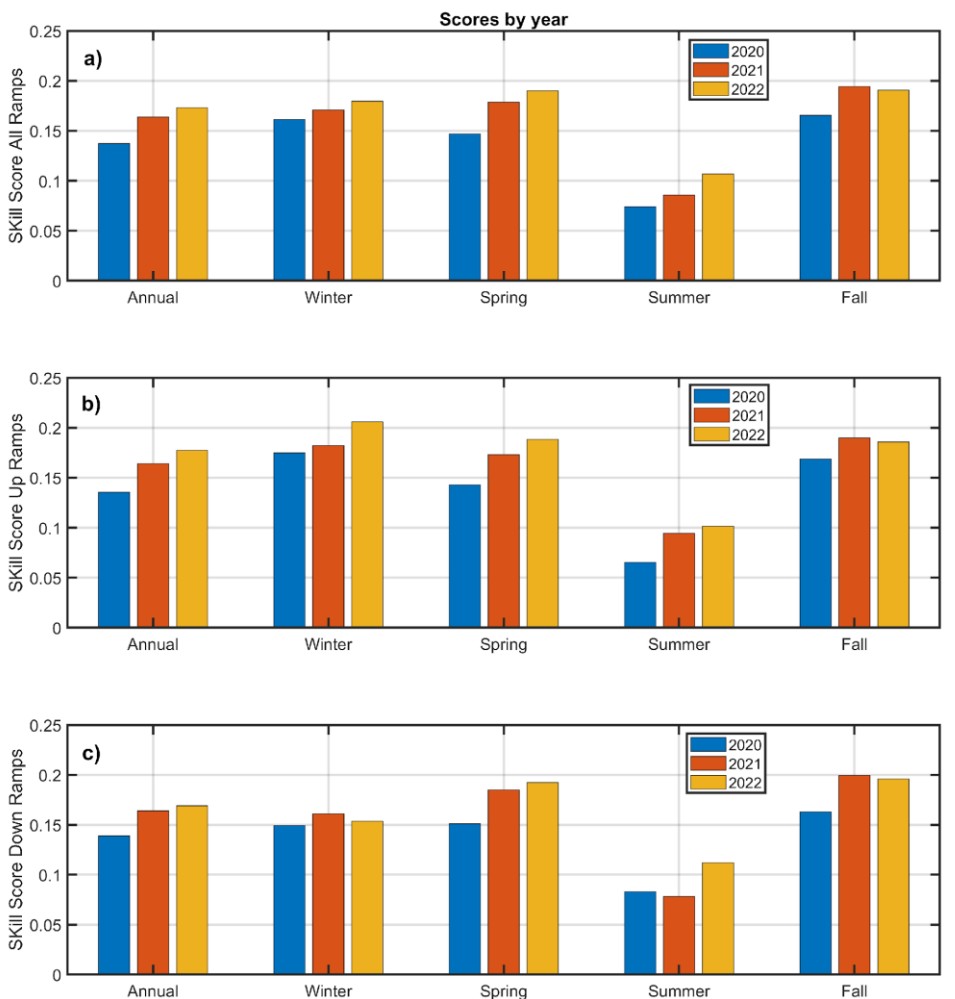

**Figure B2: Bar chart with model skill scores by years 2020, 2021, and 2022, annually and seasonally, for all ramps (panel a), up ramps only (panel b), and down ramps only (panel c).**

**Code availability**

The RT&M is publicly available online at http://www.esrl.noaa.gov/psd/products/ramp_tool/. The authors can be reached for assistance, if needed.

## Data availability

The dataset from the METeorological Aerodrome Reports (METARs) stations is available at https://aviationweather.gov/data/metar/. The United States Geological Survey (USGS) Wind Turbine database is available at https://eerscmap.usgs.gov/uswtdb/. HRRR output is available from NOAA Open Data Dissemination site at https://registry.opendata.aws/noaa-hrrr-pds/.

## Acknowledgements

We would like to acknowledge the NOAA Hollings Scholar program for supporting J. Lindblom and the NOAA / Global Systems Laboratory internship program for supporting R. Mendeke. We would like to thank Dr. Temple Lee, NOAA Air Resources Laboratory, for providing data downloading scripts and guidance to J. Lindblom as part of this project. We would also like to thank the Referees for the constructive comments. This research was supported by the NOAA cooperative agreement NA22OAR4320151, for the Cooperative Institute for Earth System Research and Data Science (CIESRDS). Additional funding for this work was provided by the NOAA Atmospheric Science for Renewable Energy (ASRE) program and by the NOAA Physical Sciences Laboratory. Data from the Southern Great Plains Central Facility, Lamont, OK were obtained from the Atmospheric Radiation Measurement Program sponsored by the U.S. Department of Energy, Office of Science, Office of Biological and Environmental Research, Climate and Environmental Sciences Division. The scientific results and conclusions, as well as any views or opinions expressed herein, are those of the authors and do not necessarily reflect those of NOAA, OAR, or the U.S. Department of Commerce.

## Author contributions

DDT is responsible for the conceptualization of the study. LB, RM, JL, and ID contributed to the formal analysis. LB, RM, JL, ID, and DDT contributed to the visualization of the results. LB and ID prepared the manuscript with writing, review and editing contributions from DDT and JMW.

## Competing interests

The authors declare that they have no conflict of interest.

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
