# Peer review of "Evaluating the ability of the operational High Resolution Rapid"

_Wind Energy Science, 2024_

## Author Comment (AC1)

**Referee #1**

**Review of "Evaluating the ability of the operational High Resolution Rapid Refresh model version 3 (HRRRv3) and version 4 (HRRRv4) to forecast wind ramp events in the US Great Plains" by Bianco et al.**

The manuscript evaluates the ability of the High Resolution Rapid Refresh (HRRR) numerical weather prediction model in forecasting wind ramp events in the U.S. Great Plains. The study focuses on two operational versions: HRRRv3 and HRRRv4. Utilizing the Ramp Tool and Metric (RT&M), it demonstrates that HRRRv4 outperforms HRRRv3 in skill, particularly in detecting up-ramp events during summer, which is vital for wind energy integration into the electric grid. The methodology includes using 10-m observational data from METAR stations and model outputs, with statistical analyses carried out for annual and seasonal variations.

The work is timely, addressing the critical need for reliable wind energy forecasting in the context of renewable energy integration. However, some areas require additional clarity or justification. The manuscript is well-written and accessible to a broad audience in meteorology and renewable energy fields.

We thank the Referee for the thoughtful and detailed comments. We hope we have addressed all of the Referee's concerns and we think that our manuscript did benefit from the constructive comments made by both Referees. In the following text, the Referee's comments are in black and our answers are in red.

**Major Comments**

1. While the manuscript provides a broad description of RT&M, it would benefit from a clearer explanation of its mathematical implementation. How is the skill score derived for different ramp characteristics (e.g., timing, amplitude, and duration)? Reference to specific equations (e.g., Bianco et al., 2016) is helpful but insufficient for readers unfamiliar with RT&M.

   More specifics on the way the skill score of the model at forecasting wind ramps is computed are included in the revised version of the manuscript (Section 2).

2. The decision to use 10 m wind speeds is justified by correlations with 80-m data and data availability. However, as acknowledged by the authors, this introduces uncertainty, particularly when converting the wind speed into power generation – small changes in wind speed can result in large changes in wind power and associated ramps. This limitation should be discussed in more detail. Since the focus of this study is the ramps, I would suggest also evaluating the ramp statistics (with wind power instead of wind speed) of those two levels to address the potential biases.

   To address both Referees' comment on the representativeness of 10 m wind speeds to evaluate model performances at 80 m agl, we included Appendix 1 to the revised

version of the manuscript. Using the HRRR output over the 2020-2022 period, we show:

- high correlation values (R = 0.84) between wind speeds at 10 and 80 m;
- high correlation (R = 0.82 for up ramps and R = 0.84 for down ramps) between the total number of modeled ramps at the METAR weather stations at these 2 levels (new Fig. A1.1);
- consistency in the normalized geographical distribution of modeled ramps between the 10 m and 80 m levels (new Fig. A1.2).

Also, although 80 m wind speeds are not measured in many locations compared to the availability of METAR stations, we used observations collected routinely at the Central Site of the ARM Observatory in OK to show high correlation between the 10 m level and the next few levels above it (R = 0.94 for 10 m vs 80 m wind speed and R = ~0.8 for 10 m vs 80 m wind power capacity factor) for all 3 years (new Fig. A1.3).

We also included some reasoning on the purpose/implications of our study/results in Section 3: *"Ramp events can be divided into those that occur because of the strong diurnal variability within the boundary layer, and those that are associated with meteorological phenomena such as cold fronts, gust fronts, or other changes in forcing from transient mesoscale pressure gradient fields. Although the diurnal variation of wind speeds at 10 m and at several 100 m can be out-of-phase (with 10 m wind speeds decreasing during the night time hours while at 300-400 m they may increase at night due to the low-level jet) diurnal variations at both heights are driven by surface and boundary layer fluxes and turbulent mixing. If improvements to the model's parameterization of those diurnal processes increases forecast skill at 10 m, one would expect that improvements to forecast skill would also be found at greater heights within the boundary layer."*

3. Both HRRRv3 and HRRRv4 have longer periods of data than what is used in this study. Why was only one year of V3 data used? Additionally, although 2021 and 2022 were both simulated by HRRRv4, the large differences observed between these two years (Figure 5,6,7, and 9) indicate that the inter-annual variation in skill may not be fully explained by the model improvements alone. This raises concerns about the representativeness of the dataset. For instance, the conclusion that there is a 50% increase in skill for summer up-ramps; how much of this improvement can be attributed to the model improvement vs inter-annual variability? Can this conclusion apply to other years? Expanding the analysis to include multiple years and evaluate the interannual variability for both versions would strengthen the conclusions.

Regarding inter-annual variability being a possible contribution to the skill of the model at forecasting wind ramps, we agree with the Referee's concern and we now mention this possibility in the main body of the manuscript (Section 5.2, discussion of Fig. 10) and also include Appendix 2 to investigate this possibility in more detail. In Appendix 2 we show that the wind speed field output at 80 m agl of the HRRR model are similar in winter months between years 2020 and 2021, but are indeed stronger

in 2022, while they are stronger in summer 2020 compared to summer months of 2021 and 2022 (new Fig. A2.1). Although there is this variability in 80 m wind speed among the years, when we look at the skill score by individual years (new Fig. A2.2), we find that while there are some differences in skill score between years 2021 and 2020 (with the same HRRRv4 model), the skill score is still improved in both years with HRRRv4 (2021 and 2022), compared to HRRRv3 (2020). This confirms that even though inter-annual variability can impact the score of the model, HRRRv4 is still doing better than HRRRv3 as previously stated.

Also, we understand the Referee's point on using a larger dataset, but this would become a much larger effort computationally, and we do believe that the addition of the Appendix discussed above helps to confirm our results.

4. For many figures, the captions are repeated in the main text. I suggest removing this redundancy to save space and instead expanding on the discussion of the figure contents.

   The figure descriptions in the main text have been shortened as suggested and moved to the figure captions.

5. The geospatial distribution of results has not been sufficiently addressed. Most statistics are averaged over all sites, and there is little discussion of the spatial variability. This aspect could be tied to the physical developments in HRRRv4. Analyzing and discussing the spatial distribution would provide additional depth to the analysis.

   Additional discussion on the geographical distribution of the results is included in the discussion of Fig. 9 (Section 5.1).

**Specific Comments**

1. Lines 37-39, please add a reference for this statement.

   Added, as suggested.

2. Lines 85-88, this paragraph seems out of place and may connect better to the paragraph starting at Line 70.

   Thanks for this suggestion, the paragraph was moved to connect with the one starting at Line 70.

3. Lines 108-109: consider moving this sentence to figure caption.

   Modified, as suggested.

4. Line 111, please also note on the rated speed.

We don't understand the suggestion of the Referee, but would be happy to include it, if clarified. Thanks.

5. Lines, please see my major comment.

   More specifics on the way the skill score of the model at forecasting wind ramps is computed are included in the revised version of the manuscript (Section 2).

6. Line 180, since METAR data is assimilated, should good performance at those locations be expected? How this result applicable to the area without METAR station available should be discussed.

   Observational datasets are assimilated at the METAR weather stations as well as everywhere else when available. The assessment of the improvement in model performances independently from the fact that the data are being assimilated, would require performing data denial experiments, which is beyond the scope of this study. A data denial study performed during the first Wind Forecast Improvement Project has been presented in Bianco et al. (2016).

   In this study, since both HRRRv3 and HRRRv4 assimilate METAR weather station observations, as well as other observations available elsewhere, we can argue that our comparison of the skill of these models at forecasting wind ramp events is still valid.

7. Lines 186-188, many down ramps occur around 00 UTC (Figure 7) when artificial "ramp" are also expected. Why not use simulation start at other hours (e.g., 6 UTC) when ramp is less frequent?

   The 00UTC down ramps are due the reduction of wind speed that happens at sunset, not to the stitching of the model forecasts.

8. Figure 5, my understanding is this figure based on model results at site locations. How about the observations? Meanwhile, please indicate that the how the size of the circles was normalized. How does this normalization influence the results?

   We agree with the Referee that Fig. 5 was difficult to interpret and reproduced it using colorbars with appropriate ranges of variability. Also, the ramp numbers presented in this figure are not normalized. The observation values are used to produce Fig. 6.

9. Line 230, larger difference between 2021 and 2022 are observed compared to their difference from 2020, suggest that the interannual variability is more important than difference in model versions?

   See comment above on the inclusion of Appendix 2 to address inter-annual variability as a possible impact on the results. As mentioned above, while there are some differences in skill score between years 2021 and 2022 (with the same HRRRv4 model) due to inter-annual variability of the wind speed field, the skill score

is still improved in both years with HRRRv4 (2021 and 2022), compared to HRRRv3 (2020). This confirms that even though inter-annual variability can impact the score of the model, HRRRv4 is still doing better than HRRRv3 as previously stated.

10. Figure 6, this figure suggests a good consistency between the years. However, the blue color spreads over a wide range of data from 0 to 100%, potentially masking large differences. Consider using more colors within the 0 to 100% range.

We agree with the Referee that Fig. 6 was difficult to interpret and we reproduced it using colorbars with appropriate ranges of variability.

11. Lines 238-240, please move this to figure caption.

Modified, as suggested.

12. Lines 245-248, Redundant with the figure caption; remove this repetition.

Modified, as suggested.

13. Figure 7, please change the title to "Diurnal variability in ramps and wind at 10 m". Visionally a better wind speed simulation in HRRRv3 compared to V4.

We agree with the Referee that HRRRv4 seems to have some bias at nighttime in wind speed. This though, does not reflect in the ramp statistic results. In this study we are looking at improvements in forecasting wind ramp events, a different metric completely from standard statistical metrics.

Also, the title of Fig. 7 has been changed, as suggested.

14. Line 257, could you elaborate how the statistics are calculated?

This statistical evaluation has been explained in more detail in the revised version of the manuscript *"Although, as discussed in Fig. 6, the number of observed ramps is in general larger than the number of model ramps, we performed a statistical analysis for the matched wind ramp events (model and observed ramps are matched when the distance between their relative central time is less than the defined time window length, i.e. 2hr for the type of ramps defined as having a ΔP/ΔT 40%/2hrs). The correlation and root mean square error (RMSE) in ΔP for these matched events at all sites are presented in Fig. 8. For HRRRv4 we used the averaged correlation coefficient and RMSE of years 2021 and 2022. With the exception of winter, both the statistical metrics improve in HRRRv4 compared to HRRRv3"*.

15. Lines 258-259, this has already been included in figure legend.

Modified, as suggested.

16. Lines 276-278, could you discussion the spatial distribution? we do see a larger improvement in the region with less ramps.

Some discussion on the geographical distribution of the results are included in the discussion of Fig. 9 (Section 5.1). Specifically, the improvement is found in all of the study area, despite the different geographical distribution of wind ramp events seen in Fig. 5.

17. Lines 281-284, already in figure caption.

Modified, as suggested.

---

## Author Comment (AC2)

**Referee #2**

The paper aims to show the modeling improvements of wind ramp events of the HRRRv4 model relative to the previous version HRRRv3. The study is well-written, highly timely, and relevant. However, the conclusions of the study, that HRRRv4 is demonstrated to generally outperform HRRRv3 for ramp events is not satisfactorily supported in the evidence provided.

I recommend major revisions to the paper before it is accepted.

We thank the Referee for the thoughtful and detailed comments. We hope we have addressed all of the Referee's concerns and we think that our manuscript did benefit from the constructive comments made by both Referees. In the following text, the Referee's comments are in black and our answers are in red.

**General comments**

- The study does not significantly quantify the influence of its assumptions. The first assumption is that model performance at 80 m is a good proxy for performance at hub height. Showing a decent overall correlation between the two model levels for the whole period is insufficient. You should at least focus on showing it for ramp events and actual observations, even if you only have a few sites available. The second assumption is that because the spatial distribution of ramp occurrence is somewhat similar in 2020 for HRRRv3 and 2021 and 2022 for HRRRv4, the comparison in performance between these different years and model versions is warranted (implied in L220-221). You should do more to show the influence of interannual variability on the results and perhaps present the results in a way that makes it easier for the reader to convince themselves of the improvements (the dots in Fig. 5 and 6 are difficult to compare).

  To address both Referees' comment on the representativeness of 10 m wind speeds to evaluate model performances at 80 m agl, we included Appendix 1 to the revised version of the manuscript. Using the HRRR output over the 2020-2022 period, we show:

  - high correlation values (R = 0.84) between wind speeds at 10 and 80 m;
  - high correlation (R = 0.82 for up ramps and R = 0.84 for down ramps) between the total number of modeled ramps at the METAR weather stations at these 2 levels (new Fig. A1.1);
  - consistency in the normalized geographical distribution of modeled ramps between the 10 m and 80 m levels (new Fig. A1.2).
  - Also, although 80 m wind speeds are not measured in many locations compared to the availability of METAR stations, we used observations collected routinely at the Central Site of the ARM Observatory in OK to show high correlation between the 10 m level and the next few levels above it (R = 0.94 for 10 m vs 80 m wind speed and R = ~0.8 for 10 m vs 80 m wind power capacity factor) for all 3 years (new Fig. A1.3).

  Regarding inter-annual variability being a possible contribution to the skill of the model at forecasting wind ramps, we agree with the Referee's concern and we now mention this possibility in the main body of the manuscript (Section 5.2, discussion of Fig. 10) and

include Appendix 2 to investigate this possibility in more detail. In Appendix 2 we show that the wind speed field output at 80 m agl of the HRRR model are similar in winter months between years 2020 and 2021, but are indeed stronger in 2022, while they are stronger in summer 2020 compared to summer months of 2021 and 2022 (new Fig. A2.1). Although there is this variability in 80 m wind speed among the years, when we look at the skill score by individual years (new Fig. A2.2), we notice that while there are some differences in skill score between years 2021 and 2020 (with the same HRRRv4 model), the skill score is still improved in both years with HRRRv4 (2021 and 2022), compared to HRRRv3 (2020). This confirms that even though inter-annual variability can impact the score of the model, HRRRv4 is still doing better than HRRRv3 as previously stated.

We also included some reasoning on the purpose/implications of our study/results in Section 3: *"Ramp events can be divided into those that occur because of the strong diurnal variability within the boundary layer, and those that are associated with meteorological phenomena such as cold fronts, gust fronts, or other changes in forcing from transient mesoscale pressure gradient fields. Although the diurnal variation of wind speeds at 10 m and at several 100 m can be out-of-phase (with 10 m wind speeds decreasing during the night time hours while at 300-400 m they may increase at night due to the low-level jet) diurnal variations at both heights are driven by surface and boundary layer fluxes and turbulent mixing. If improvements to the model's parameterization of those diurnal processes increases forecast skill at 10 m, one would expect that improvements to forecast skill would also be found at greater heights within the boundary layer."*

We agree with the Referee that Fig. 5 and 6 were difficult to interpret and we modified both of them using colorbars with appropriate ranges of variability.

- Alternative hypotheses that could explain the results (the performance improvements seen), such as natural variability, are not discussed or tested, weakening the results and conclusions drawn. Given that HRRRv3 and HRRRv4 are compared across different periods, at the least, some effort must be made to rule out natural variability as the driver of differences. L230 of the paper could indicate that interannual variability is not negligible.

  We agree with the Referees' comments and added Appendix 2 to discuss the possible impact of inter-annual variability to the results. See comments above on inter-annual variability for more details on the content of Appendix 2.

- Please sure you are following the guidelines of the journal regarding notation, dates, math symbols, etc.: https://www.wind-energy-science.net/submission.html#math, e.g., "1700 UTC" -> "17:00 UTC", avoid hyphens with abbreviated units (e.g. "10-m wind"), and many more cases.

  We tried to follow the notation guidelines in the revised version of the manuscript.

**Specific comments**

- L100-101: The RT&M method is so central to this study that you should spell out the details here, not simply refer to another paper

  More specifics on the way the skill score of the model at forecasting wind ramps is computed are included in the revised version of the manuscript (Section 2).

- Figure 2: I suggest indicating the study area on the map(s). In general, Fig. 2 is presented but not discussed much. Perhaps you can relate the mean and standard deviation to the number of ramp events experienced. Perhaps you could even make a ramp occurrence map.

  As suggested, we included a box indicating the study area on panels a and b of Fig. 2. Some features revealed by this figure are now discussed in the text, when discussing Fig. 2, and referred to later discussion in the manuscript. Ramp occurrence maps are already presented in Fig. 5 (model) and 6 (model/obs) for the study area.

- L169-170: Please state how the temporal interpolation was done

  We have reworded this statement in the revised version of the manuscript to: *"we have linearly interpolated the METAR observations in time to the HRRR output times"*.

- L189-190: Please explain the 3-point smoother in more detail. Is it simply the average between the two? Or something else?

  As suggested, we have provided a description of that 3-point filter in Section 3.2 (*"i.e., the model output valid at 23:00 was the weighted average of the output valid at 22:00, 23:00, and 00:00 with the two outer points having 25% weight and the central time having a 50% weight, whereas the model output valid at 00:00 was the weighted average of the output valid at 23:00, 00:00, and 01:00 with the same weighting approach"*).

- Figure 4: please add the runs for the 2021-04-07 00Z and 2021-04-11 00Z initializations to the figure to allow the reader to follow the source for the red line throughout the period

  Thanks for the suggestion. Lines relative to 2021-04-07 00Z and 2021-04-12 00Z initialization times have been added to the figure, as requested.

- Figure 5: How much of the spatial variation in ramp events is explained by the variation in mean wind speed?

  Text referring to the fact that the geographical distribution of the number of wind ramp events presented in Fig. 5 agrees with the annual wind speed geographical distribution presented in Fig. 2 has now been added to the discussion of Fig. 5.

- Figures 5 and 6: it would be helpful to show the frequency distributions of all the samples. This would also help the reader see more clearly the improvements you mention

  We have included many figures to the revised manuscript and we hope they are helping better characterize the improvements we mention.

- Figure 7: It would be valuable if you reflected on how these diurnal cycles may look different for typical hub height. For example, you mention the importance of low-level jets in the text. How would they change the picture? One could, perhaps, expect a reverse cycle at higher altitudes. Also, I would suggest using local time-of-day values or indicating the typical ranges corresponding to day- and nighttime.

  As the Referee points out, at higher heights we could indeed expect a reverse cycle in the diurnal cycle of wind speed. This consideration is particularly valid at the height of the nose of the LLJ. This has been added to the revised version of the manuscript.

  Also, sunrise and sunset times have been included in the updated version of Fig. 7, as suggested.

- "Newmann" -> "Newman" in three places on page 6

  Thanks. Corrected.

- Small suggestion: your author contributions are very short and general/vague. You can take a look at https://publications.copernicus.org/services/contributor_roles_taxonomy.html and perhaps make it more specific

  The Authors' contribution section has been expanded as requested.

---

## Referee Report (RR1)

The paper aims to show the modeling improvements of wind ramp events of the HRRRv4 model relative to the previous version HRRRv3. The study is well-written, highly timely, and relevant. However, the conclusions of the study, that HRRRv4 is demonstrated to generally outperform HRRRv3 for ramp events is not satisfactorily supported in the evidence provided.

I recommend major revisions to the paper before it is accepted.

We thank the Referee for the thoughtful and detailed comments. We hope we have addressed all of the Referee's concerns and we think that our manuscript did benefit from the constructive comments made by both Referees. In the following text, the Referee's comments are in black and our answers are in red.

Thank you for the thorough revision and your replies to my comments. My new replies are added in blue.

You have addressed many of my comments and added valuable new insights supporting the paper's conclusions. However, the main critiques are still not fully addressed.

Although the interannual variability is better described now, the attribution to interannual variability in the reported improvements is still unclear. For instance, you mention a 50% improvement in skillfulness in detecting up-ramps in the summer with HRRRv4 vs. HRRRv3, but how much can be attributed to natural variability?

You have added valuable reflections regarding using 10 m data to make conclusions for typical hub height levels but are not sufficiently considering the uncertainty that comes from this. For example, you write that "If improvements to the model's parameterization of those diurnal processes increases forecast skill at 10 m, one would expect that improvements to forecast skill would also be found at greater heights within the boundary layer." - which is plausible, but by no means guaranteed. I don't think you have provided enough evidence to, e.g., rule out that LLJ-induced ramps at higher altitudes are significant or show whether HRRRv4 is better at forecasting these than HRRRv3. You have provided strong indications that this is the case, but not hard evidence.

**General comments**

- The study does not significantly quantify the influence of its assumptions. The first assumption is that model performance at 10 m is a good proxy for performance at hub height. Showing a decent overall correlation between the two model levels for the whole period is insufficient. You should at least focus on showing it for ramp events and actual observations, even if you only have a few sites available. The second assumption is that because the spatial distribution of ramp occurrence is somewhat similar in 2020 for HRRRv3 and 2021 and 2022 for HRRRv4, the comparison in performance between these different years and model versions is warranted (implied in L220-221). You should do more to show the influence of interannual variability on the results, and perhaps

present the results in a way that makes it easier for the reader to convince themselves of this (the dots in Fig. 5 and 6 are difficult to compare).

To address both Referees' comment on the representativeness of 10 m wind speeds to evaluate model performances at 80 m agl, we included Appendix 1 to the revised version of the manuscript. Using the HRRR output over the 2020-2022 period, we show:

- high correlation values (R = 0.84) between wind speeds at 10 and 80 m;
- high correlation (R = 0.82 for up ramps and R = 0.84 for down ramps) between the total number of modeled ramps at the METAR weather stations at these 2 levels (new Fig. A1.1);
- consistency in the normalized geographical distribution of modeled ramps between the 10 m and 80 m levels (new Fig. A1.2).
- Also, although 80 m wind speeds are not measured in many locations compared to the availability of METAR stations, we used observations collected routinely at the Central Site of the ARM Observatory in OK to show high correlation between the 10 m level and the next few levels above it (R = 0.94 for 10 m vs 80 m wind speed and R = ~0.8 for 10 m vs 80 m wind power capacity factor) for all 3 years (new Fig. A1.3).

Thank you for adding this. It is very helpful. However, it also affirms that approx. 30% of the model's variance in ramp events at 80 m is unexplained by ramps at 10 m. Add that to the fact that the modeled ramps at 10 m explain only about 40% of the variance in the observations (based on the annual value in Fig. 8); it possibly leaves quite a bit of variance in actual ramps at 80 m unaccounted for by your results from 10 m. Your conclusions should be expressed with this uncertainty in mind.

Regarding inter-annual variability being a possible contribution to the skill of the model at forecasting wind ramps, we agree with the Referee's concern and we now mention this possibility in the main body of the manuscript (Section 5.2, discussion of Fig. 10) and include Appendix 2 to investigate this possibility in more detail. In Appendix 2 we show that the wind speed field output at 80 m agl of the HRRR model are similar in winter months between years 2020 and 2021, but are indeed stronger in 2022, while they are stronger in summer 2020 compared to summer months of 2021 and 2022 (new Fig. A2.1). Although there is this variability in 80 m wind speed among the years, when we look at the skill score by individual years (new Fig. A2.2), we notice that while there are some differences in skill score between years 2021 and 2020 (with the same HRRRv4 model), the skill score is still improved in both years with HRRRv4 (2021 and 2022), compared to HRRRv3 (2020). This confirms that even though inter-annual variability can impact the score of the model, HRRRv4 is still doing better than HRRRv3 as previously stated.

These additional plots are much more convincing than those in the main text, indicating consistent improvement from HRRRv3 to HRRRv4 (improvement in 8 out of 8 seasons for ramp-ups and 7 out of 8 for ramps-downs). I would elevate them to the main text.

That said, whether each year is statistically average or an outlier is still unclear. A more extended dataset (e.g., re-analysis) could indicate this.

We also included some reasoning on the purpose/implications of our study/results in Section 3: "Ramp events can be divided into those that occur because of the strong diurnal variability within the boundary layer, and those that are associated with meteorological phenomena such as cold fronts, gust fronts, or other changes in forcing from transient mesoscale pressure gradient fields. Although the diurnal variation of wind speeds at 10 m and at several 100 m can be out-of-phase (with 10 m wind speeds decreasing during the night time hours while at 300-400 m they may increase at night due to the low-level jet) diurnal variations at both heights are driven by surface and boundary layer fluxes and turbulent mixing. If improvements to the model's parameterization of those diurnal processes increases forecast skill at 10 m, one would expect that improvements to forecast skill would also be found at greater heights within the boundary layer."

I appreciate this added consideration; it is an important one. The last part here is speculation. While it is plausible that improvements at 10 m coincide with improvements higher up, it cannot be taken as given.

We agree with the Referee that Fig. 5 and 6 were difficult to interpret and we modified both of them using colorbars with appropriate ranges of variability.

Thank you for this. It can still be challenging to see where improvements were made vs not. I would change the colors to reflect instead the improvements in the ratio model/obs from 2020 to 2021 and 2022, respectively.

- Alternative hypotheses that could explain the results, such as natural variability, are not discussed or tested, weakening the results and conclusions drawn. Given that HRRRv3 and HRRRv4 are compared across different periods, at the least, some effort must be made to rule out natural variability as the driver of differences.

  We agree with the Referees' comments and added Appendix 2 to discuss the possible impact of inter-annual variability to the results. See comments above on inter-annual variability for more details on the content of Appendix 2.

  See the comment above.

- Please sure you are following the guidelines of the journal regarding notation, dates, math symbols, etc.: https://www.wind-energy-science.net/submission.html#math, e.g., "1700 UTC" -> "17:00 UTC", avoid hyphens with abbreviated units (e.g. "10-m wind"), and many more cases.

  We tried to follow the notation guidelines in the revised version of the manuscript.

**Specific comments**

- L100-101: The RT&M method is so central to this study that you should spell out the details here, not simply refer to another paper

  More specifics on the way the skill score of the model at forecasting wind ramps is computed are included in the revised version of the manuscript (Section 2).

  Thank you.

- Figure 2: I would suggest indicating the US states on the maps. In general, the figure is presented but discussed much. Perhaps relate the mean and standard deviation to the number of ramp events experienced. Perhaps you could even make a ramp occurrence map.

  As suggested, we included a box indicating the study area on panels a and b of Fig. 2. Some features revealed by this figure are now discussed in the text, when discussing Fig.
  2, and referred to later discussion in the manuscript. Ramp occurrence maps are already presented in Fig. 5 (model) and 6 (model/obs) for the study area.

  The ramp occurrence maps in Fig. 5 and 6 are at the tower locations. A ramp map from the model (based on each grid cell) and the association between ramps and the mean and standard deviation and wind speed would be valuable, e.g., for assessing the influence of interannual variability in wind speed on ramp-occurrence.

- L169-170: Please state how the temporal interpolation was done

  We have reworded this statement in the revised version of the manuscript to: "we have linearly interpolated the METAR observations in time to the HRRR output times".

  Thank you.

- L189-190: Please explain the 3-point smoother in more detail. Is it simply the average between the two? Or something else?

  As suggested, we have provided a description of that 3-point filter in Section 3.2 ("i.e., the
  model output valid at 23:00 was the weighted average of the output valid at 22:00, 23:00,
  and 00:00 with the two outer points having 25% weight and the central time having a 50%
  weight, whereas the model output valid at 00:00 was the weighted average of the output valid at 23:00, 00:00, and 01:00 with the same weighting approach").

  Thank you.

- Figure 4: please add the runs for the 2021-04-07 00Z and 2021-04-11 00Z initializations to the figure to allow the reader to follow the source for the red line throughout the period

  Thanks for the suggestion. Lines relative to 2021-04-07 00Z and 2021-04-12 00Z initialization times have been added to the figure, as requested.

  Thank you.

- Figure 5: How much of the spatial variation in ramp events is explained by the variation in mean wind speed?

  Text referring to the fact that the geographical distribution of the number of wind ramp events presented in Fig. 5 agrees with the annual wind speed geographical distribution presented in Fig. 2 has now been added to the discussion of Fig. 5.

  Good addition, but how much is explained by the annual wind speed? E.g., what is the correlation between ramps and annual wind speed?

- Figures 5 and 6: it would be helpful to show the frequency distributions of all the samples. This would also help the reader see more clearly the improvements you mention

  We have included many figures to the revised manuscript and we hope they are helping better characterize the improvements we mention.

  Yes, but most new figures are maps or simple box plots. What I meant above was that it would be helpful if you plotted the frequency distribution of ramp errors (e.g., model/obs or model-obs) for all the towers, with each year making up different distributions. Aggregate improvements would become clearer in such a figure.

- Figure 7: It would be valuable if you reflected on how these diurnal cycles may look different for typical hub height. For example, you mention the importance of low-level jets in the text. How would they change the picture? One could, perhaps, expect a reverse cycle at higher altitudes. Also, I would suggest using local time-of-day values or indicating the typical ranges corresponding to day- and nighttime.

  As the Referee points out, at higher heights we could indeed expect a reverse cycle in the diurnal cycle of wind speed. This consideration is particularly valid at the height of the
  nose of the LLJ. This has been added to the revised version of the manuscript.
  Also, sunrise and sunset times have been included in the updated version of Fig. 7, as
  Suggested.

  Much appreciated.

- "Newmann" -> "Newman" in three places on page 6

Thanks. Corrected.

- Small suggestion: your author contributions are very short and general/vague. You can take a look at https://publications.copernicus.org/services/contributor_roles_taxonomy.html and perhaps make it more specific

The Authors' contribution section has been expanded as requested.

---

## Referee Report (RR2)

Dear authors, thank you for the changes. I think it has improved the manuscript, and you have addressed my points adequately. So, although some uncertainty remains around the influence of interannual variability on the annual ramp statistics, and whether the results generalize to hub-heights, I find the paper in its current form a valuable study showing improvements to ramp forecasting made in an important operational NWP model, and therefore recommend that the editor accept your paper. A few remarks below, all in **magenta.**

We thank the Referee for the additional comments provided to our manuscript. We hope we have addressed all of the Referee's concerns and we think that our manuscript did benefit from the constructive comments made by both Referees in both rounds of the review process. Please note that, while the red text corresponds to our original replies to the Referee, we have marked the answers to the most recent comments by the Referee **in green**, in the text below.

The paper aims to show the modeling improvements of wind ramp events of the HRRRv4 model relative to the previous version HRRRv3. The study is well-written, highly timely, and relevant. However, the conclusions of the study, that HRRRv4 is demonstrated to generally outperform HRRRv3 for ramp events is not satisfactorily supported in the evidence provided.

I recommend major revisions to the paper before it is accepted.

We thank the Referee for the thoughtful and detailed comments. We hope we have addressed all of the Referee's concerns and we think that our manuscript did benefit from the constructive comments made by both Referees. In the following text, the Referee's comments are in black and our answers are in red.

Thank you for the thorough revision and your replies to my comments. My new replies are added in blue.

You have addressed many of my comments and added valuable new insights supporting the paper's conclusions. However, the main critiques are still not fully addressed.

Although the interannual variability is better described now, the attribution to interannual variability in the reported improvements is still unclear. For instance, you mention a 50% improvement in skillfulness in detecting up-ramps in the summer with HRRRv4 vs. HRRRv3, but how much can be attributed to natural variability?

You have added valuable reflections regarding using 10 m data to make conclusions for typical hub height levels but are not sufficiently considering the uncertainty that comes from this. For example, you write that "If improvements to the model's parameterization of those diurnal processes increases forecast skill at 10 m, one would expect that improvements to forecast skill would also be found at greater heights within the boundary layer." - which is plausible, but by no means guaranteed. I don't think you have provided enough evidence to, e.g., rule out that LLJ-induced ramps at higher altitudes are significant or show whether HRRRv4 is better at

forecasting these than HRRRv3. You have provided strong indications that this is the case, but not hard evidence.

We again thank the Referee for the additional comments provided to our manuscript. We have marked the answers to the Referee in green, in the text below.

**General comments**

- The study does not significantly quantify the influence of its assumptions. The first assumption is that model performance at 10 m is a good proxy for performance at hub height. Showing a decent overall correlation between the two model levels for the whole period is insufficient. You should at least focus on showing it for ramp events and actual observations, even if you only have a few sites available. The second assumption is that because the spatial distribution of ramp occurrence is somewhat similar in 2020 for HRRRv3 and 2021 and 2022 for HRRRv4, the comparison in performance between these different years and model versions is warranted (implied in L220-221). You should do more to show the influence of interannual variability on the results, and perhaps present the results in a way that makes it easier for the reader to convince themselves of this (the dots in Fig. 5 and 6 are difficult to compare).

  To address both Referees' comment on the representativeness of 10 m wind speeds to evaluate model performances at 80 m agl, we included Appendix 1 to the revised version of the manuscript. Using the HRRR output over the 2020-2022 period, we show:

  - high correlation values (R = 0.84) between wind speeds at 10 and 80 m;
  - high correlation (R = 0.82 for up ramps and R = 0.84 for down ramps) between the total number of modeled ramps at the METAR weather stations at these 2 levels (new Fig. A1.1);
  - consistency in the normalized geographical distribution of modeled ramps between the 10 m and 80 m levels (new Fig. A1.2).
  - Also, although 80 m wind speeds are not measured in many locations compared to the availability of METAR stations, we used observations collected routinely at the Central Site of the ARM Observatory in OK to show high correlation between the 10 m level and the next few levels above it (R = 0.94 for 10 m vs 80 m wind speed and R = ~0.8 for 10 m vs 80 m wind power capacity factor) for all 3 years (new Fig. A1.3).

  Thank you for adding this. It is very helpful. However, it also affirms that approx. 30% of the model's variance in ramp events at 80 m is unexplained by ramps at 10 m. Add that to the fact that the modeled ramps at 10 m explain only about 40% of the variance in the observations (based on the annual value in Fig. 8); it possibly leaves quite a bit of variance in actual ramps at 80 m unaccounted for by your results from 10 m. Your conclusions should be expressed with this uncertainty in mind.

We agree with the Referee on this remark. We expressed this uncertainty in our results, more clearly in the revised version of the manuscript, including additional text as suggested. In Appendix A: "We recognize that a correlation of 0.84 explains only 70% of the variance between 10 and 80 m wind speeds and number of ramps at those two heights. The remaining 30% are uncertainties that could possibly reflect different diurnal wind speed and ramp events behaviours at these two heights."

This is a good addition, but I would have preferred to see it reflected also in the main text/conclusions.

Regarding inter-annual variability being a possible contribution to the skill of the model at forecasting wind ramps, we agree with the Referee's concern and we now mention this possibility in the main body of the manuscript (Section 5.2, discussion of Fig. 10) and include Appendix 2 to investigate this possibility in more detail. In Appendix 2 we show that the wind speed field output at 80 m agl of the HRRR model are similar in winter months between years 2020 and 2021, but are indeed stronger in 2022, while they are stronger in summer 2020 compared to summer months of 2021 and 2022 (new Fig. A2.1). Although there is this variability in 80 m wind speed among the years, when we look at the skill score by individual years (new Fig. A2.2), we notice that while there are some differences in skill score between years 2021 and 2020 (with the same HRRRv4 model), the skill score is still improved in both years with HRRRv4 (2021 and 2022), compared to HRRRv3 (2020). This confirms that even though inter-annual variability can impact the score of the model, HRRRv4 is still doing better than HRRRv3 as previously stated.

These additional plots are much more convincing than those in the main text, indicating consistent improvement from HRRRv3 to HRRRv4 (improvement in 8 out of 8 seasons for ramp-ups and 7 out of 8 for ramps-downs). I would elevate them to the main text.

We appreciate your suggestion on elevating Appendix B to the body of the manuscript, but in our opinion, it would break the focus on the final results of the main analysis, that the HRRRv4 skill at forecasting wind ramp events improved in respect to the HRRRv3 annually and by season, which is presented in Fig. 10. We believe that the inclusion of Appendix B after the first round of revisions was indeed valuable in showing that although there might be some inter-annual variability, the HRRRv4 is doing better than the HRRRv3 in two consecutive years, validating our results.

That said, whether each year is statistically average or an outlier is still unclear. A more extended dataset (e.g., re-analysis) could indicate this.

Thanks for the suggestion, but using reanalysis is out of the scope of our study. A completely new study could be performed using other datasets, as for instance reanalysis, but it would be a completely different approach. In this study we wanted to use real observations.

We also included some reasoning on the purpose/implications of our study/results in Section 3: "Ramp events can be divided into those that occur because of the strong diurnal variability within the boundary layer, and those that are associated with meteorological phenomena such as cold fronts, gust fronts, or other changes in forcing from transient mesoscale pressure gradient fields. Although the diurnal variation of wind speeds at 10 m and at several 100 m can be out-of-phase (with 10 m wind speeds decreasing during the night time hours while at 300-400 m they may increase at night due to the low-level jet) diurnal variations at both heights are driven by surface and boundary layer fluxes and turbulent mixing. If improvements to the model's parameterization of those diurnal processes increases forecast skill at 10 m, one would expect that improvements to forecast skill would also be found at greater heights within the boundary layer."

I appreciate this added consideration; it is an important one. The last part here is speculation. While it is plausible that improvements at 10 m coincide with improvements higher up, it cannot be taken as given.

We agree with the Referee that the last part of the sentence is speculation and we clarified it in the revised version of the manuscript (instead of "one would expect that …" we now say "one could speculate that …"). Moreover, in Appendix A, we included some analysis at the SGP ARM site: "Additionally, at this site we computed the correlation between the model and the radiosonde observed winds at 80 m for those three years, finding an improvement in R from 0.85 in 2020 (HRRRv3), to 0.86 in 2021 and 2022 (HRRRv4). We also used high-frequency (10 Hz) observations of wind speed from a sonic anemometer (R3-50, manufactured by Gill Instruments) located on a 60 m tower at the same site. Sonic data were averaged at the top of the hour (plus/minus 5 minutes) providing a more complete dataset compared to the radiosonde one. In this case we found an improvement in R from 0.78 in 2020 (HRRRv3), to 0.79 in 2021 (HRRRv4), to 0.84 in 2022 (HRRRv4) between 80 m model and 60 m sonic wind observations. Furthermore, the comparison with the 60 m sonic observations was repeated dividing the dataset into night time and daytime, similarly to what was presented in Section 5.3. For daytime, correlation coefficient values were found to be equal to 0.84 in 2020 (HRRRv3), to 0.80 in 2021 (HRRRv4), and to 0.87 in 2022 (HRRRv4). For night time, correlation coefficient values were found to be equal to 0.73 in 2020 (HRRRv3), to 0.78 in 2021 (HRRRv4), and to 0.81 in 2022 (HRRRv4). Although this is at one site only, this result aligns with the findings presented in Section 5.3, that in stable conditions the correlation was much improved in HRRRV4 relative to HRRRV3. This supports our speculation that improvements of HRRRv4 compared to HRRRv3 to ramp skill at 10 m would also be found at hub height, although to prove this statement with more certainty, we would need a more appropriate dataset."

Although one could argue that you should further group the ramps into seasons, I believe you have quite thoroughly shown that ramp forecasts improved between HRRRv3 and v4, broadly at 10 m, and higher up at the SGP ARM site. Some uncertainty

remains about the influence of interannual variability, but we have already covered that part previously.

We agree with the Referee that Fig. 5 and 6 were difficult to interpret and we modified both of them using colorbars with appropriate ranges of variability.

Thank you for this. It can still be challenging to see where improvements were made vs not. I would change the colors to reflect instead the improvements in the ratio model/obs from 2020 to 2021 and 2022, respectively.

We thank the Referee for this valuable suggestion. We included 2 additional panels to Fig. 6. In panel d we present improvements in the ratio mod/obs for HRRRv4 (2021) vs HRRRv3 (2020), and in panel e for HRRRv4 (2022) vs HRRRv3 (2020). This inclusion shows that the ratio does in fact improve in the HRRRv4 version of the model at most of the stations for both years, proving our point more clearly. The following text was added to the revised version of the manuscript, before Fig. 6 "To further show that the ratio between the number of forecast wind ramps and those observed improves over the years and the model versions, we present the geographical distribution of the improvement from 2020 to 2021 and from 2020 to 2022, in panels d and e of Fig. 6, respectively. As noticeable, at most of the stations (72.5% of panel d, and 67% of panel e) the improvement is positive."

Thank you. This makes it clearer.

- Alternative hypotheses that could explain the results, such as natural variability, are not discussed or tested, weakening the results and conclusions drawn. Given that HRRRv3 and HRRRv4 are compared across different periods, at the least, some effort must be made to rule out natural variability as the driver of differences.

  We agree with the Referees' comments and added Appendix 2 to discuss the possible impact of inter-annual variability to the results. See comments above on inter-annual variability for more details on the content of Appendix 2.

  See the comment above.

  See comment above on our decision to not move Appendix B to the main body of the text.

- Please sure you are following the guidelines of the journal regarding notation, dates, math symbols, etc.: https://www.wind-energy-science.net/submission.html#math, e.g., "1700 UTC" -> "17:00 UTC", avoid hyphens with abbreviated units (e.g. "10-m wind"), and many more cases.

  We tried to follow the notation guidelines in the revised version of the manuscript.

**Specific comments**

- L100-101: The RT&M method is so central to this study that you should spell out the details here, not simply refer to another paper

  More specifics on the way the skill score of the model at forecasting wind ramps is computed are included in the revised version of the manuscript (Section 2).

  Thank you.

- Figure 2: I would suggest indicating the US states on the maps. In general, the figure is presented but discussed much. Perhaps relate the mean and standard deviation to the number of ramp events experienced. Perhaps you could even make a ramp occurrence map.

  As suggested, we included a box indicating the study area on panels a and b of Fig. 2. Some features revealed by this figure are now discussed in the text, when discussing Fig.
  2, and referred to later discussion in the manuscript. Ramp occurrence maps are already presented in Fig. 5 (model) and 6 (model/obs) for the study area.

  The ramp occurrence maps in Fig. 5 and 6 are at the tower locations. A ramp map from the model (based on each grid cell) and the association between ramps and the mean and standard deviation and wind speed would be valuable, e.g., for assessing the influence of interannual variability in wind speed on ramp-occurrence.

  This would require an incredible effort of data analysis and we think that this would not add much to the goal of this study. Nevertheless, we believe that the map in Fig. 5 shows clearly the geographical distribution of ramps over the study area.

- L169-170: Please state how the temporal interpolation was done

  We have reworded this statement in the revised version of the manuscript to: "we have linearly interpolated the METAR observations in time to the HRRR output times".

  Thank you.

- L189-190: Please explain the 3-point smoother in more detail. Is it simply the average between the two? Or something else?

  As suggested, we have provided a description of that 3-point filter in Section 3.2 ("i.e., the
  model output valid at 23:00 was the weighted average of the output valid at 22:00, 23:00,
  and 00:00 with the two outer points having 25% weight and the central time having a 50%
  weight, whereas the model output valid at 00:00 was the weighted average of the output

valid at 23:00, 00:00, and 01:00 with the same weighting approach”).

Thank you.

- Figure 4: please add the runs for the 2021-04-07 00Z and 2021-04-11 00Z initializations to the figure to allow the reader to follow the source for the red line throughout the period

  Thanks for the suggestion. Lines relative to 2021-04-07 00Z and 2021-04-12 00Z initialization times have been added to the figure, as requested.

  Thank you.

- Figure 5: How much of the spatial variation in ramp events is explained by the variation in mean wind speed?

  Text referring to the fact that the geographical distribution of the number of wind ramp events presented in Fig. 5 agrees with the annual wind speed geographical distribution presented in Fig. 2 has now been added to the discussion of Fig. 5.

  Good addition, but how much is explained by the annual wind speed? E.g., what is the correlation between ramps and annual wind speed?

  To answer this question, we computed the correlation between the number of ramps and the annual averaged wind speed over all stations. We found values on the order of 0.8 for all years. However, we don't believe that this adds much to our results because the wind speed itself is not completely representative for ramp analysis. For instance, higher wind speeds do not necessarily reflect in more ramp events, as the wind power capacity might be completely saturated at high wind speeds. What is important in ramp analysis is the rapid variation of wind speed over a short period of time and this is why a new metric (RT&M) was created.

- Figures 5 and 6: it would be helpful to show the frequency distributions of all the samples. This would also help the reader see more clearly the improvements you mention

  We have included many figures to the revised manuscript and we hope they are helping better characterize the improvements we mention.

  Yes, but most new figures are maps or simple box plots. What I meant above was that it would be helpful if you plotted the frequency distribution of ramp errors (e.g., model/obs or model-obs) for all the towers, with each year making up different distributions. Aggregate improvements would become clearer in such a figure.

  The RT&M provides the skill of the model at forecasting ramps. This skill is a combination of 3 possible errors, one in Central Time, one in Delta Power, and one in

Delta Time. The error improvement (larger skill values for HRRRv4 compared to HRRRv3) is already included in Fig. 9 for each tower location.

- Figure 7: It would be valuable if you reflected on how these diurnal cycles may look different for typical hub height. For example, you mention the importance of low-level jets in the text. How would they change the picture? One could, perhaps, expect a reverse cycle at higher altitudes. Also, I would suggest using local time-of-day values or indicating the typical ranges corresponding to day- and nighttime.

  As the Referee points out, at higher heights we could indeed expect a reverse cycle in the diurnal cycle of wind speed. This consideration is particularly valid at the height of the
  nose of the LLJ. This has been added to the revised version of the manuscript.
  Also, sunrise and sunset times have been included in the updated version of Fig. 7, as
  Suggested.

  Much appreciated.

- "Newmann" -> "Newman" in three places on page 6

  Thanks. Corrected.

- Small suggestion: your author contributions are very short and general/vague. You can take a look at https://publications.copernicus.org/services/contributor_roles_taxonomy.html and perhaps make it more specific

  The Authors' contribution section has been expanded as requested.

---

## Referee Report (RR3)

[referee-annotated manuscript omitted]

---

## Author Response (AR2)

**Referee #1**

We thank the Referee for the additional comments provided to our manuscript. We hope we have addressed all of the Referee's concerns and we think that our manuscript did benefit from the constructive comments made by both Referees in both rounds of the review process. We have marked the answers to the Referee **in green**, in the text below.

I appreciate the authors' efforts in addressing my initial comments, particularly the extended analyses in Appendices 1 and 2, providing additional insight into both the 10 m vs. 80 m representativeness question and the role of interannual variability.

Thanks for the positive feedback.

1. Although Appendix 1 helps justify the reliance on 10 m data given the limited availability of 80 m observations, I believe the focus should not only be on justifying this decision, but also on acknowledging the uncertainty it introduces. The authors' analyses indicate correlations near 0.8 between wind speeds at 10 m and 80 m, which may be viewed as relatively high yet potentially limiting under certain atmospheric conditions. For instance overnight stable layers can cause significant divergence between near-surface and hub-height winds. It would be especially helpful for readers to know under which conditions conclusions drawn from 10 m data are most robust, and under which conditions further caution is needed.

We thank the Referee for this good suggestion. To explore how our conclusions are robust under different atmospheric conditions (for instance stable vs unstable), we divided the dataset into nighttime and daytime. We did this as there were no observations at multiple levels to determine stability in the lowest portion of the troposphere. We then recomputed the models' skills and skill improvements for daytime vs nighttime for ramps defined as $\Delta P/\Delta T \geq 40\%/2hrs$. As suggested by the Referee, we wanted to see if the improvements we presented in our study were still consistent between these different atmospheric conditions.

The daytime period is selected to be 12:00 to 22:00 UTC and the nighttime is 23:00 UTC plus 00:00 to 11:00 UTC. The results of this exercise are presented in the table below.

| Year | 2020 (HRRRv3) | 2021 (HRRRv4) | 2022 (HRRRv4) |
|---|---|---|---|
| Model Skill daytime | 0.234938 | 0.259056 | 0.256345 |
| Model Skill nighttime | 0.192997 | 0.210443 | 0.235195 |
| Daytime Skill improvement compared to 2020 | | 10.2659 % | 9.11194 % |
| Nighttime Skill improvement compared to 2020 | | 9.03930 % | 21.8645 % |

For both HRRRv4 years (2021 and 2022), we find an improvement in the skill of the model at forecasting ramp events of the order of ~10% compared to 2020 (HRRRv3). For the nighttime, we actually find a twice higher improvement (>20%) for year 2022 compared to year 2020.

These results show that, although there might be differences in values, the improvements are still consistently positive for both HRRRv4 years (and both daytime and nighttime) compared to the HRRRv3 year. Results of this additional analysis are included in the revised version of the manuscript (new Section 5.3).

Also, additional analysis using a sonic anemometer mounted on a 60 m tower at the SGP ARM site was added to Appendix A: *"We also used high-frequency (10 Hz) observations of wind speed from a sonic anemometer (R3-50, manufactured by Gill Instruments) located on a 60 m tower at the same site. Sonic data were averaged at the top of the hour (plus/minus 5 minutes) providing a more complete dataset compared to the radiosonde one. In this case we found an improvement in R from 0.78 in 2020 (HRRRv3), to 0.79 in 2021 (HRRRv4), to 0.84 in 2022 (HRRRv4) between 80 m model and 60 m sonic wind observations. Furthermore, the comparison with the 60 m sonic observations was repeated dividing the dataset into night time and daytime, similarly to what was presented in Section 5.3. For daytime, correlation coefficient values were found to be equal to 0.84 in 2020 (HRRRv3), to 0.80 in 2021 (HRRRv4), and to 0.87 in 2022 (HRRRv4). For night time, correlation coefficient values were found to be equal to 0.73 in 2020 (HRRRv3), to 0.78 in 2021 (HRRRv4), and to 0.81 in 2022 (HRRRv4). Although this is at one site only, this result aligns with the findings presented in Section 5.3, that in stable conditions the correlation was much improved in HRRRV4 relative to HRRRV3. This supports our speculation that improvements of HRRRv4 compared to HRRRv3 to ramp skill at 10 m would also be found at hub height, although to prove this statement with more certainty, we would need a more appropriate dataset."*

Quantitatively demonstrating how the evaluation uncertainty varies with weather conditions, such as stable, unstable, or low-level jet events, or other ways defining the weather, would show how reliable these findings are across different environmental scenarios.

As discussed in the answer to the comment above, we did take the Referee's comment under consideration and we investigated if the improvements reported in the study were consistent between stable vs unstable conditions. We found that this is the case.

On the other hand, differentiating among other different weather conditions, such as for instance low-level-jet events, frontal passages… is unfortunately beyond our possibilities with the current dataset. Nevertheless, we agree with the Referee that this would be an interesting way to perform an analysis on the skill of the model at forecasting ramp events. We are considering performing such an evaluation over a different dataset, now being collected, where different weather conditions are crucial for ramp analysis (frontal passages, LLJ, fog, sea breeze…). For that dataset weather conditions and predominant atmospheric conditions are recorded daily in an event log and this will give us the possibility to perform a similar analysis to the one presented in our study, but also adding the possibility to measure the skill of the model under different atmospheric conditions.

2. Regarding the concern about interannual variability, Appendix 2 addresses some aspects of how year-to-year differences influence the results. The authors show that HRRRv4 demonstrates skill improvements for both 2021 and 2022 relative to 2020's HRRRv3 period, which is encouraging. However, the differences between 2021 and 2022 themselves remain notable. My main question is how much of the observed skill differences should be attributed to the model upgrades versus the interannual variability. One way to clarify this issue would be to compare ramp forecasts during different meteorological scenarios (for example, days with

strong frontal passages, days without strong forcing, or nights with identified low-level jet conditions). If HRRRv4 outperforms HRRRv3 across a variety of these scenarios, it would strengthen the argument that the improvements are indeed model-driven.

Please, see reply to the comment above about a future analysis analyzing the skill of the model as a function of different atmospheric phenomena.

**Referee #2**

We thank the Referee for the additional comments provided to our manuscript. We hope we have addressed all of the Referee's concerns and we think that our manuscript did benefit from the constructive comments made by both Referees in both rounds of the review process. Please note that, while the red text corresponds to our original replies to the Referee, we have marked the answers to the most recent comments by the Referee **in green**, in the text below.

The paper aims to show the modeling improvements of wind ramp events of the HRRRv4 model relative to the previous version HRRRv3. The study is well-written, highly timely, and relevant.
However, the conclusions of the study, that HRRRv4 is demonstrated to generally outperform HRRRv3 for ramp events is not satisfactorily supported in the evidence provided.

I recommend major revisions to the paper before it is accepted.

We thank the Referee for the thoughtful and detailed comments. We hope we have addressed all of the Referee's concerns and we think that our manuscript did benefit from the constructive comments made by both Referees. In the following text, the Referee's comments are in black and our answers are in red.

Thank you for the thorough revision and your replies to my comments. My new replies are added in blue.

You have addressed many of my comments and added valuable new insights supporting the paper's conclusions. However, the main critiques are still not fully addressed.

Although the interannual variability is better described now, the attribution to interannual variability in the reported improvements is still unclear. For instance, you mention a 50% improvement in skillfulness in detecting up-ramps in the summer with HRRRv4 vs. HRRRv3, but how much can be attributed to natural variability?

You have added valuable reflections regarding using 10 m data to make conclusions for typical hub height levels but are not sufficiently considering the uncertainty that comes from this. For example, you write that "If improvements to the model's parameterization of those diurnal processes increases forecast skill at 10 m, one would expect that improvements to forecast skill would also be found at greater heights within the boundary layer." - which is plausible, but by no means guaranteed. I don't think you have provided enough evidence to, e.g., rule out that LLJ-induced ramps at higher altitudes are significant or show whether HRRRv4 is better at forecasting these than HRRRv3. You have provided strong indications that this is the case, but not hard evidence.

We again thank the Referee for the additional comments provided to our manuscript. We have

marked the answers to the Referee in green, in the text below.

**General comments**

- The study does not significantly quantify the influence of its assumptions. The first assumption is that model performance at 10 m is a good proxy for performance at hub height. Showing a decent overall correlation between the two model levels for the whole period is insufficient. You should at least focus on showing it for ramp events and actual observations, even if you only have a few sites available. The second assumption is that because the spatial distribution of ramp occurrence is somewhat similar in 2020 for HRRRv3 and 2021 and 2022 for HRRRv4, the comparison in performance between these different years and model versions is warranted (implied in L220-221). You should do more to show the influence of interannual variability on the results, and perhaps the results in a way that makes it easier for the reader to convince themselves of this (the dots in Fig. 5 and 6 are difficult to compare).

To address both Referees' comment on the representativeness of 10 m wind speeds to evaluate model performances at 80 m agl, we included Appendix 1 to the revised version of the manuscript. Using the HRRR output over the 2020-2022 period, we show:

- high correlation values (R = 0.84) between wind speeds at 10 and 80 m;
- high correlation (R = 0.82 for up ramps and R = 0.84 for down ramps) between the total number of modeled ramps at the METAR weather stations at these 2 levels (new Fig. A1.1);
- consistency in the normalized geographical distribution of modeled ramps between the 10 m and 80 m levels (new Fig. A1.2).
- Also, although 80 m wind speeds are not measured in many locations compared to the availability of METAR stations, we used observations collected routinely at the Central Site of the ARM Observatory in OK to show high correlation between the 10 m level and the next few levels above it (R = 0.94 for 10 m vs 80 m wind speed and R = ~0.8 for 10 m vs 80 m wind power capacity factor) for all 3 years (new Fig. A1.3).

Thank you for adding this. It is very helpful. However, it also affirms that approx. 30% of the model's variance in ramp events at 80 m is unexplained by ramps at 10 m. Add that to the fact that the modeled ramps at 10 m explain only about 40% of the variance in the observations (based on the annual value in Fig. 8); it possibly leaves quite a bit of variance in actual ramps at 80 m unaccounted for by your results from 10 m. Your conclusions should be expressed with this uncertainty in mind.

We agree with the Referee on this remark. We expressed this uncertainty in our results, more clearly in the revised version of the manuscript, including additional text as suggested. In Appendix A: *"We recognize that a correlation of 0.84 explains only 70% of the variance between 10 and 80 m wind speeds and number of ramps at those two heights. The remaining 30% are uncertainties that could possibly reflect different diurnal*

*wind speed and ramp events behaviours at these two heights."*

Regarding inter-annual variability being a possible contribution to the skill of the model at forecasting wind ramps, we agree with the Referee's concern and we now mention this possibility in the main body of the manuscript (Section 5.2, discussion of Fig. 10) and include Appendix 2 to investigate this possibility in more detail. In Appendix 2 we show that the wind speed field output at 80 m agl of the HRRR model are similar in winter months between years 2020 and 2021, but are indeed stronger in 2022, while they are stronger in summer 2020 compared to summer months of 2021 and 2022 (new Fig. A2.1). Although there is this variability in 80 m wind speed among the years, when we look at the skill score by individual years (new Fig. A2.2), we notice that while there are some differences in skill score between years 2021 and 2020 (with the same HRRRv4 model), the skill score is still improved in both years with HRRRv4 (2021 and 2022), compared to HRRRv3 (2020). This confirms that even though inter-annual variability can impact the score of the model, HRRRv4 is still doing better than HRRRv3 as previously stated.

These additional plots are much more convincing than those in the main text, indicating consistent improvement from HRRRv3 to HRRRv4 (improvement in 8 out of 8 seasons for ramp-ups and 7 out of 8 for ramps-downs). I would elevate them to the main text.

We appreciate your suggestion on elevating Appendix B to the body of the manuscript, but in our opinion, it would break the focus on the final results of the main analysis, that the HRRRv4 skill at forecasting wind ramp events improved in respect to the HRRRv3 annually and by season, which is presented in Fig. 10. We believe that the inclusion of Appendix B after the first round of revisions was indeed valuable in showing that although there might be some inter-annual variability, the HRRRv4 is doing better than the HRRRv3 in two consecutive years, validating our results.

That said, whether each year is statistically average or an outlier is still unclear. A more extended dataset (e.g., re-analysis) could indicate this.

Thanks for the suggestion, but using reanalysis is out of the scope of our study. A completely new study could be performed using other datasets, as for instance reanalysis, but it would be a completely different approach. In this study we wanted to use real observations.

We also included some reasoning on the purpose/implications of our study/results in Section 3: "Ramp events can be divided into those that occur because of the strong diurnal variability within the boundary layer, and those that are associated with meteorological phenomena such as cold fronts, gust fronts, or other changes in forcing from transient mesoscale pressure gradient fields. Although the diurnal variation of wind speeds at 10 m and at several 100 m can be out-of-phase (with 10 m wind speeds decreasing during the night time hours while at 300-400 m they may increase at night due to the low-level jet) diurnal variations at both heights are driven by surface and boundary layer fluxes and turbulent mixing. If improvements

to the model's parameterization of those diurnal processes increases forecast skill at 10 m, one would expect that improvements to forecast skill would also be found at greater heights within the boundary layer."

I appreciate this added consideration; it is an important one. The last part here is speculation. While it is plausible that improvements at 10 m coincide with improvements higher up, it cannot be taken as given.

We agree with the Referee that the last part of the sentence is speculation and we clarified it in the revised version of the manuscript (instead of *"one would expect that …"* we now say *"one could speculate that …"*). Moreover, in Appendix A, we included some analysis at the SGP ARM site: *"Additionally, at this site we computed the correlation between the model and the radiosonde observed winds at 80 m for those three years, finding an improvement in R from 0.85 in 2020 (HRRRv3), to 0.86 in 2021 and 2022 (HRRRv4). We also used high-frequency (10 Hz) observations of wind speed from a sonic anemometer (R3-50, manufactured by Gill Instruments) located on a 60 m tower at the same site. Sonic data were averaged at the top of the hour (plus/minus 5 minutes) providing a more complete dataset compared to the radiosonde one. In this case we found an improvement in R from 0.78 in 2020 (HRRRv3), to 0.79 in 2021 (HRRRv4), to 0.84 in 2022 (HRRRv4) between 80 m model and 60 m sonic wind observations. Furthermore, the comparison with the 60 m sonic observations was repeated dividing the dataset into night time and daytime, similarly to what was presented in Section 5.3. For daytime, correlation coefficient values were found to be equal to 0.84 in 2020 (HRRRv3), to 0.80 in 2021 (HRRRv4), and to 0.87 in 2022 (HRRRv4). For night time, correlation coefficient values were found to be equal to 0.73 in 2020 (HRRRv3), to 0.78 in 2021 (HRRRv4), and to 0.81 in 2022 (HRRRv4). Although this is at one site only, this result aligns with the findings presented in Section 5.3, that in stable conditions the correlation was much improved in HRRRV4 relative to HRRRV3. This supports our speculation that improvements of HRRRv4 compared to HRRRv3 to ramp skill at 10 m would also be found at hub height, although to prove this statement with more certainty, we would need a more appropriate dataset."*

We agree with the Referee that Fig. 5 and 6 were difficult to interpret and we modified both of them using colorbars with appropriate ranges of variability.

Thank you for this. It can still be challenging to see where improvements were made vs not. I would change the colors to reflect instead the improvements in the ratio model/obs from 2020 to 2021 and 2022, respectively.

We thank the Referee for this valuable suggestion. We included 2 additional panels to Fig. 6. In panel d we present improvements in the ratio mod/obs for HRRRv4 (2021) vs HRRRv3 (2020), and in panel e for HRRRv4 (2022) vs HRRRv3 (2020).
This inclusion shows that the ratio does in fact improve in the HRRRv4 version of the model at most of the stations for both years, proving our point more clearly. The following text was added to the revised version of the manuscript, before Fig. 6 *"To further show that the ratio between the number of forecast wind ramps and those observed improves over the years and the model versions, we present the geographical distribution of the improvement from 2020 to 2021 and from 2020 to 2022, in panels d and e of Fig. 6, respectively. As noticeable, at most of the stations (72.5% of panel d,*

*and 67% of panel e) the improvement is positive."*

- Alternative hypotheses that could explain the results, such as natural variability, are not discussed or tested, weakening the results and conclusions drawn. Given that HRRRv3 and HRRRv4 are compared across different periods, at the least, some effort must be made to rule out natural variability as the driver of differences.

  We agree with the Referees' comments and added Appendix 2 to discuss the possible impact of inter-annual variability to the results. See comments above on inter-annual variability for more details on the content of Appendix 2.

  See the comment above.

  See comment above on our decision to not move Appendix B to the main body of the text.

- Please sure you are following the guidelines of the journal regarding notation, dates, math symbols, etc.: https://www.wind-energy-science.net/submission.html#math, e.g., "1700 UTC" -> "17:00 UTC", avoid hyphens with abbreviated units (e.g. "10-m wind"), and many more cases.

  We tried to follow the notation guidelines in the revised version of the manuscript.

**Specific comments**

- L100-101: The RT&M method is so central to this study that you should spell out the details here, not simply refer to another paper

  More specifics on the way the skill score of the model at forecasting wind ramps is computed are included in the revised version of the manuscript (Section 2).

  Thank you.

- Figure 2: I would suggest indicating the US states on the maps. In general, the figure is presented but discussed much. Perhaps relate the mean and standard deviation to the number of ramp events experienced. Perhaps you could even make a ramp occurrence map.

  As suggested, we included a box indicating the study area on panels a and b of Fig. 2. Some features revealed by this figure are now discussed in the text, when discussing Fig. 2, and referred to later discussion in the manuscript. Ramp occurrence maps are already presented in Fig. 5 (model) and 6 (model/obs) for the study area.

  The ramp occurrence maps in Fig. 5 and 6 are at the tower locations. A ramp map from the model (based on each grid cell) and the association between ramps and the mean and standard deviation and wind speed would be valuable, e.g., for assessing the influence of interannual variability in wind speed on ramp-occurrence.

This would require an incredible effort of data analysis and we think that this would not add much to the goal of this study. Nevertheless, we believe that the map in Fig. 5 shows clearly the geographical distribution of ramps over the study area.

- L169-170: Please state how the temporal interpolation was done

We have reworded this statement in the revised version of the manuscript to: "we have linearly interpolated the METAR observations in time to the HRRR output times".

Thank you.

- L189-190: Please explain the 3-point smoother in more detail. Is it simply the average between the two? Or something else?

As suggested, we have provided a description of that 3-point filter in Section 3.2 ("i.e., the model output valid at 23:00 was the weighted average of the output valid at 22:00, 23:00, and 00:00 with the two outer points having 25% weight and the central time having a 50% weight, whereas the model output valid at 00:00 was the weighted average of the output valid at 23:00, 00:00, and 01:00 with the same weighting approach").

Thank you.

- Figure 4: please add the runs for the 2021-04-07 00Z and 2021-04-11 00Z initializations to the figure to allow the reader to follow the source for the red line throughout the period

Thanks for the suggestion. Lines relative to 2021-04-07 00Z and 2021-04-12 00Z initialization times have been added to the figure, as requested.

Thank you.

- Figure 5: How much of the spatial variation in ramp events is explained by the variation in mean wind speed?

Text referring to the fact that the geographical distribution of the number of wind ramp events presented in Fig. 5 agrees with the annual wind speed geographical distribution presented in Fig. 2 has now been added to the discussion of Fig. 5.

Good addition, but how much is explained by the annual wind speed? E.g., what is the correlation between ramps and annual wind speed?

To answer this question, we computed the correlation between the number of ramps and the annual averaged wind speed over all stations. We found values on the order of 0.8 for all years. However, we don't believe that this adds much to our results because the wind speed itself is not completely representative for ramp analysis. For instance, higher wind speeds do not necessarily reflect in more ramp events, as the wind power capacity might be completely saturated at high wind speeds. What is important in ramp analysis

is the rapid variation of wind speed over a short period of time and this is why a new metric (RT&M) was created.

- Figures 5 and 6: it would be helpful to show the frequency distributions of all the samples. This would also help the reader see more clearly the improvements you mention

  We have included many figures to the revised manuscript and we hope they are helping better characterize the improvements we mention.

  Yes, but most new figures are maps or simple box plots. What I meant above was that it would be helpful if you plotted the frequency distribution of ramp errors (e.g., model/obs or model-obs) for all the towers, with each year making up different distributions. Aggregate improvements would become clearer in such a figure.

  The RT&M provides the skill of the model at forecasting ramps. This skill is a combination of 3 possible errors, one in Central Time, one in Delta Power, and one in Delta Time. The error improvement (larger skill values for HRRRv4 compared to HRRRv3) is already included in Fig. 9 for each tower location.

- Figure 7: It would be valuable if you reflected on how these diurnal cycles may look different for typical hub height. For example, you mention the importance of low-level jets in the text. How would they change the picture? One could, perhaps, expect a reverse cycle at higher altitudes. Also, I would suggest using local time-of-day values or indicating the typical ranges corresponding to day- and nighttime.

  As the Referee points out, at higher heights we could indeed expect a reverse cycle in the diurnal cycle of wind speed. This consideration is particularly valid at the height of the nose of the LLJ. This has been added to the revised version of the manuscript. Also, sunrise and sunset times have been included in the updated version of Fig. 7, as suggested.

  Much appreciated.

- "Newmann" -> "Newman" in three places on page

  Thanks. Corrected.

- Small suggestion: your author contributions are very short and general/vague. You can take a look at https://publications.copernicus.org/services/contributor_roles_taxonomy.html and perhaps make it more specific

  The Authors' contribution section has been expanded as requested.